# Miga-mediated endoplasmic reticulum–mitochondria contact sites regulate neuronal homeostasis

Lingna Xu[1][†], Xi Wang[1][†], Jia Zhou[1], Yunyi Qiu[1], Weina Shang[1], Jun-Ping Liu[2], Liquan Wang[3], Chao Tong[1,2,3]*

[1]MOE Key Laboratory for Biosystems Homeostasis & Protection and Innovation Center for Cell Signaling Network, Life Sciences Institute, Zhejiang University, Hangzhou, China; [2]Institute of Ageing Research, Hangzhou Normal University College of Medicine, Hangzhou, China; [3]The Second Affiliated Hospital, School of Medicine, Zhejiang University, Hangzhou, China

**Abstract** Endoplasmic reticulum (ER)–mitochondria contact sites (ERMCSs) are crucial for multiple cellular processes such as calcium signaling, lipid transport, and mitochondrial dynamics. However, the molecular organization, functions, regulation of ERMCS, and the physiological roles of altered ERMCSs are not fully understood in higher eukaryotes. We found that Miga, a mitochondrion located protein, markedly increases ERMCSs and causes severe neurodegeneration upon overexpression in fly eyes. Miga interacts with an ER protein Vap33 through its FFAT-like motif and an amyotrophic lateral sclerosis (ALS) disease related Vap33 mutation considerably reduces its interaction with Miga. Multiple serine residues inside and near the Miga FFAT motif were phosphorylated, which is required for its interaction with Vap33 and Miga-mediated ERMCS formation. The interaction between Vap33 and Miga promoted further phosphorylation of upstream serine/threonine clusters, which fine-tuned Miga activity. Protein kinases CKI and CaMKII contribute to Miga hyperphosphorylation. MIGA2, encoded by the *miga* mammalian ortholog, has conserved functions in mammalian cells. We propose a model that shows Miga interacts with Vap33 to mediate ERMCSs and excessive ERMCSs lead to neurodegeneration.

*For correspondence:
ctong@zju.edu.cn

[†]These authors contributed equally to this work

Competing interests: The authors declare that no competing interests exist.

## Introduction

Intracellular organelles with membranes are characteristic features of eukaryotic cells. The compartmentalization of the cytoplasm enables specific cellular activities occur in restricted regions. The physical contacts between different organelles at defined loci allow both material and information exchange between organelles (*Phillips and Voeltz, 2016*; *Eisenberg-Bord et al., 2016*). ER–mitochondria contact sites (ERMCSs) are the most studied organelle contacts, which have been observed repeatedly in ultrastructural electron microscopy (*Morré et al., 1971*; *Franke and Kartenbeck, 1971*) and subcellular fractionation studies (*Pickett et al., 1980*; *Meier et al., 1981*).

ERMCSs play various roles including phospholipid exchange and calcium flux between ER and mitochondria (*Krols et al., 2016*). ERMCSs also function as platforms for mitochondrial division (*Friedman et al., 2011*), coupling mitochondrial DNA synthesis with mitochondrial fission (*Lewis et al., 2016*), autophagosome formation (*Hamasaki et al., 2013*), and mitophagy (*Böckler and Westermann, 2014*). Although ERMCSs changes have been reported in Alzheimer's disease (*Schreiner et al., 2015*; *Hedskog et al., 2013*) and metabolic diseases such as obesity (*Arruda et al., 2014*), there is no direct evidence indicating that defective ERMCSs led to disease conditions. The physiological consequences of defective ERMCSs were not clear.

The tethering molecules mediating ERMCSs in yeast form a protein complex called ER–mitochondria encounter structure (ERMES) which contains the outer membrane mitochondrial proteins Mdm10 and Mdm34, cytosolic protein Mdm12, and integral ER protein Mmm1 (*Lang et al., 2015*). The ER protein complex EMC (*Lahiri et al., 2014*; *Voss et al., 2012*) and Ltc1/Lam6 (*Murley et al., 2015*; *Elbaz-Alon et al., 2015*) have also been reported to regulate contacts between ER and mitochondria. In mammalian cells, multiple proteins have been reported to tether ER and mitochondria. Mitochondrial voltage-dependent anion-selective channel protein 1 (VDAC1) and the ER $Ca^{2+}$ channel IP3R are a pair of molecules mediating ERMCSs in mammals (*Szabadkai et al., 2006*). Although debatable, MFN2, the mitofusin that is required for mitochondrial outer membrane fusion, has been reported to tether ER and mitochondria through homotypic interactions (*Naon et al., 2016*). ER protein VAPB and the mitochondrial protein PTPIP51 bind to each other to tether the two organelles (*Stoica et al., 2014*). Recent studies in flies revealed that Miro, a mitochondrial protein containing two GTPase domains and two $Ca^{2+}$-sensing EF hand domains, regulates ERMCS (*Lee et al., 2016*; *Lee et al., 2018*). In addition, proteins such as PACS2 (*Simmen et al., 2005*), Drp1 (*Friedman et al., 2011*), and Atg14 (*Hamasaki et al., 2013*) have also been reported to be localized to the ERMCSs. The loss of single pair of tethering molecules usually has mild effects to ERMCS, which is likely due to redundancy. The components and the organization of ERMCS are still not fully understood.

Many studies showed that ERMCSs are dynamic structures that undergo active remodeling upon different stimuli (*Yang et al., 2018*; *Valm et al., 2017*; *Guo et al., 2018*). However, it is not clear how the tethering molecules are regulated in response to different cellular needs.

Our previous study identified a family of mitochondrial proteins called Mitoguardin (Miga) (*Zhang et al., 2016*). Miga contains a function unknown domain that is highly conserved from *Caenorhabditis elegans* to humans. Miga localizes on the outer membrane of mitochondria and promotes mitochondrial fusion through interaction with mitoPLD, a divergent family member of the phospholipase D family that is required for the fusion of mitochondrial outer membranes (*Huang et al., 2011*). Loss of *Miga* leads to fragmented mitochondria and reduced mitochondrial functions. Flies lacking *Miga* do not grow beyond the early pupal stage. The mosaic eyes in flies with *Miga* mutant clones degenerate with aging. Mammals have two Miga proteins: MIGA1 and MIGA2. Mice with single or double knockout (KO) of *Miga1* and *Miga2* are viable. Both *Miga2* and *Miga1/2* KO mice showed reduced body weight and body fat. Under high-fat diet consumption, *Miga2* KO mice has minimal lipidosis (*Podrini et al., 2015*). In a combined association study, an SNP located in *Miga1* was shown to be associated with subscapular skin-fold thickness in humans and back fat thickness in pigs (*Lee et al., 2011*). In addition, the female *Miga1/2* KO mice were shown to be sub-fertile with reduced ovarian activity, oocyte quality, and embryo developmental potentials (*Liu et al., 2017*; *Liu et al., 2016*). *Miga2* KO mice also has immune defects and showed severe depression (*Fan et al., 2019*; *Gao et al., 2017*). A most recent study showed that MIGA2 binds VAP proteins in ER and is involved in forming contacts between mitochondria, the ER, and lipid droplets. They also demonstrated that MIGA2 is required for adipocyte differentiation (*Freyre et al., 2019*).

In this study, we found that Miga interact with Vap33 to mediate ERMCSs. Miga is phosphorylated by casein kinase I (CKI) and $Ca^{2+}$/calmodulin-dependent protein kinase II (CaMKII) at multiple sites, which affect the interaction between Miga and Vap33 and fine-tuned Miga activity. The extra ERMCSs caused by overexpression of Miga or other tethering molecules led to severe neurodegeneration, which might shed light on the molecular mechanisms underlying neurodegenerative diseases such as Alzheimer's disease.

## Results

### Miga overexpression led to increased ERMCS and severe retinal degeneration

Our previous studies indicated that a mitochondrial outer membrane protein Miga is required for neuronal homeostasis (*Zhang et al., 2016*). Miga forms a complex with MitoPLD, which promotes mitochondrial fusion through regulating mitochondrial membrane lipid composition. Interestingly, we found that Miga overexpression led to severe retinal degeneration in fly eyes. We used *GMR-*

*Gal4*, a Gal4 driver expressing in the developing eyes, to drive *UAS-Miga* expression and examined the adult fly eyes at days 1 and 30. The adult eyes with Miga overexpression looked grossly normal from outside. However, when we examine the retina with the transmission electron microscopy (TEM) analysis (*Figure 1A–D*), we found that both the rhabdomere numbers and sizes were greatly reduced in the Miga overexpressing fly eyes, and the reductions were progressively enhanced with aging (*Figure 1I–L*).

We wondered whether the retinal degeneration of the Miga overexpressing eyes was due to the increased mitochondrial fusion. Therefore, we overexpressed Marf, the *Drosophila* mitofusin (*Sandoval et al., 2014*), and MitoPLD in a same manner as we overexpressed Miga (*Figure 1E–H*). Surprisingly, neither Marf nor MitoPLD overexpression reduced the rhabdomere numbers or sizes in 1-day-old flies. Overexpression of Marf, but not MitoPLD, reduced the rhabdomere numbers in 30 days old flies, but the degeneration phenotype was much milder than that in Miga overexpressing eyes (*Figure 1I–L*).

We carefully examined the retina in 1-day-old flies and saw many circular shaped membrane structures inside the photoreceptor cells. Some of the circular, membraned structures were found to be donut-shaped mitochondria with closely attached ER tubules (*Figure 1C'*). There were also regular shaped mitochondria with close contacts with ER (*Figure 1C'*), which were seldom seen in wild-type control eyes (*Figure 1A'*). The proximity between ER and mitochondria at the contact sites were close to 10 nm and the ribosomes were excluded (*Figure 1C'*). Although several studies suggested that MFN2, the mammalian Marf homolog, mediated ERMCSs (*Phillips and Voeltz, 2016*; *de Brito and Scorrano, 2008*), we did not observe any increase in ERMCS when Marf was overexpressed in the fly eyes (*Figure 1E'*, 1F'). MitoPLD overexpression did not affect ERMCSs either (*Figure 1G'*, 1H'). These data suggested that Miga might have a function to establish ERMCSs and extra ERMCSs led to neurodegeneration.

## Miga forms complex with Vap33 and mediates ERMCSs

To understand Miga function, we performed tandem immunoprecipitation (IP) in the cultured S2 cells with FLAG-HA tandem tagged overexpressed Miga and examined its binding partner with mass spectrometry (*Figure 2—source data 4*). The ER protein Vap33 was one of the binding partners of Miga. The mammalian Vap33 orthologs are VAPA and VAPB. Both proteins are ER proteins mediating contacts between ER and other organelles. Point mutation in VAPB has been identified in amyotrophic lateral sclerosis (ALS) patients (*Nishimura et al., 2004*; *Kabashi et al., 2013*). Recent study in mammalian cells found that MIGA2 form a complex with VAP proteins (*Freyre et al., 2019*). We confirmed the interaction between Vap33 and Miga by IP. Miga and Vap33 could pull down each other in both directions. Interestingly, Vap33$^{P58S}$, the ALS disease mimicking Vap33 mutation, had lower affinity to Miga than its wild type (*Figure 2A*).

VAP proteins interact with its partner through an FFAT motif EFFDAXE (*Murphy and Levine, 2016*). Miga has a conserved sequence that is similar to the FFAT motif but with two acidic amino acids changed to serine. We prepared an FFAT motif mutant form of Miga (Miga$^{FM}$) with the 247$^{th}$ phenylalanine and the 249$^{th}$ serine residues changed to alanine (*Figure 2B*). Miga$^{FM}$ failed to bind to Vap33 (*Figure 2C*). We expressed a genomic rescue construct of Miga (*Zhang et al., 2016*) that contains the genomic fragment of Miga and a 3 × HA tag fused at the C-terminus just before the stop codon of Miga and a Flag tagged Vap33 in S2 cells and performed IP experiments. Since the genomic rescue construct of Miga contains the regulatory sequences from Miga locus, it likely expresses Miga in an endogenous level. Indeed, the level Miga-HA was much less than the level of overexpressed ones. We found that Miga and Vap33 could pull down each other in this condition (*Figure 2D*). We expressed both wildtype and mutant Miga with Vap33 in fly fat body tissues. Vap33 had a diffused pattern when it was expressed alone (*Figure 2E*). Both Miga and Miga$^{FM}$ showed mitochondrial patterns when they were expressed alone (*Figure 2F and H*). When Vap33 and Miga were co-expressed, Vap33 was recruited by Miga and the two proteins were co-localized (*Figure 2G*). However, Miga$^{FM}$ expression failed to recruit Vap33 and its own pattern did not change upon Vap33 expression (*Figure 2I*). To test whether Miga overexpression indeed could increase ERMCSs, we examined the fly fat body tissues with TEM. In the wildtype larvae fat body tissues, around 12% mitochondria had contacts with ER (*Figure 2J and Q*) and ERMCS length occupied about 12% of the mitochondrial perimeter (*Figure 2J and R*). The average distance between ER and mitochondria at the ERMCSs is around 22 nm (*Figure 2J and S*). Vap33 overexpression alone

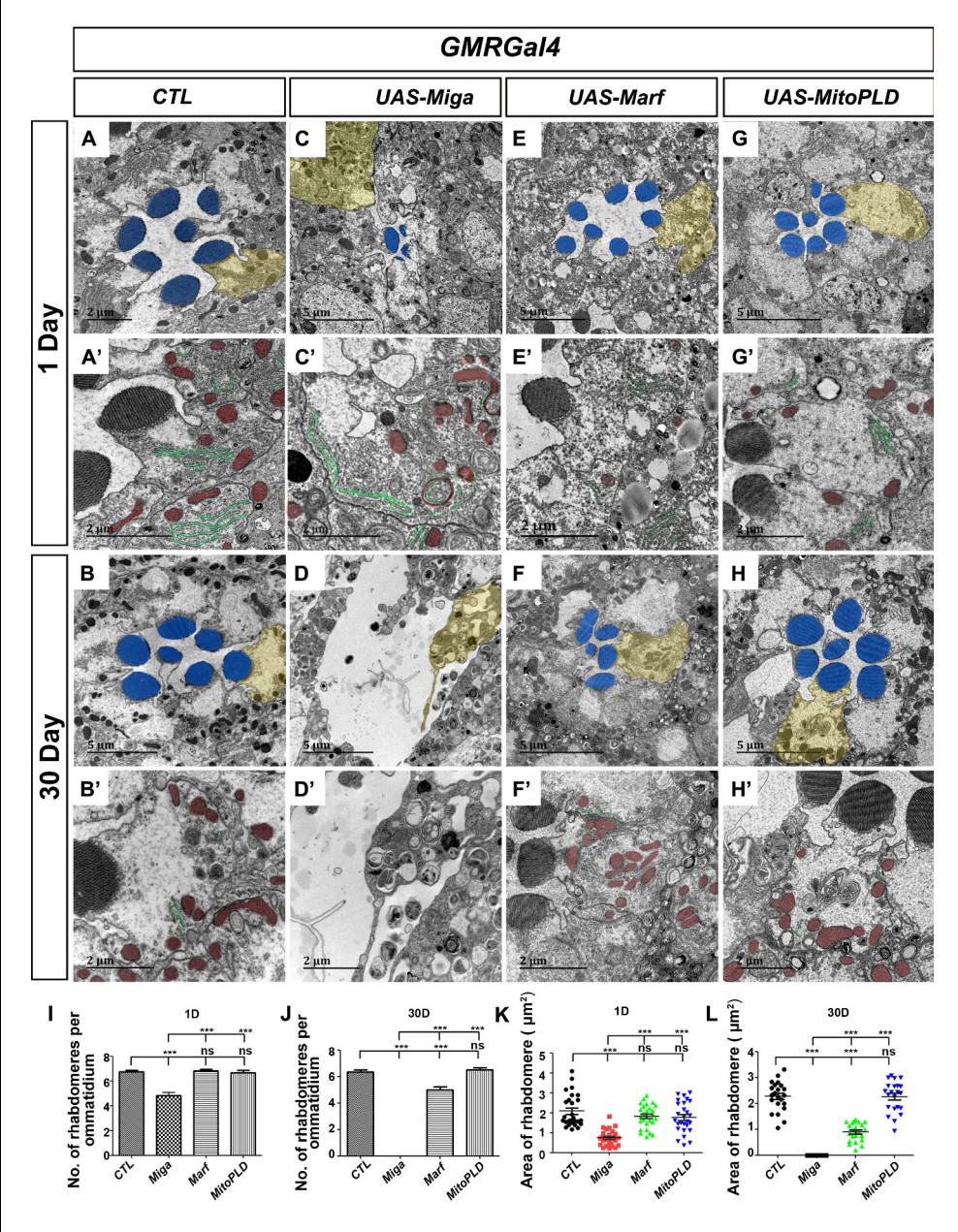

**Figure 1.** Miga overexpression led to severe retinal degeneration. TEM analysis was performed for the retina thin sections of young (1 Day) and old (30 Day) flies with indicated genotypes. (**A–B'**) The ommatidia of the control (CTL) flies showed seven photoreceptor cells with intact rhabdomeres (highlighted with blue pseudo-color) at both the young (**A, A'**) and old (**B, B'**) stages. (**C–D'**) *GMR-Gal4* driven Miga overexpression resulted in reduction of rhabdomere number and size in the 1-day-old flies (**C, C'**) and the loss of photoreceptor cells in 30-day-old flies (**D, D'**). Large amount of circular membrane structures accumulated in the photoreceptor cells. ER tubules (green) attached to the mitochondria (red) were increased in the photoreceptor cells (**C'**). (**E–F**) *GMR-Gal4*-driven Marf overexpression did not change the number and size of rhabdomeres in the 1-day-old flies (**E, E'**), but slightly reduced the number and size of rhabdomeres in the 30-day-old animals (**F, F'**). (**G–H'**) *GMR-Gal4*-driven MitoPLD overexpression did not affect the number and size of rhabdomeres in both 1-day and 30-day-old animals. (**A'–H'**) are enlarged views of the photoreceptor cells highlighted with yellow pseudo-color in (**A–H**). The ER tubules was marked with green pseudo-color and the mitochondria was marked by the red pseudo-color. (**I**) Quantification of the rhabdomere numbers per ommatidia in the 1-day-old flies with indicated genotypes. n = 12 for each genotype, data are represented as mean + SD. ns, not significant; ***, p<0.001; one-way ANOVA/Bonferroni's multiple comparisons test. (**J**) Quantification of the rhabdomere numbers per ommatidia in the 30-day-old flies

*Figure 1 continued on next page*

*Figure 1 continued*

with indicated genotypes. n = 12 for each genotype, data are represented as mean + SD. ns, not significant; ***, p<0.001; one-way ANOVA/Bonferroni's multiple comparisons test. (K) Quantification of the rhabdomere size in the 1-day-old flies with indicated genotypes. n = 27 for each genotype, data are represented as mean + SD. ns, not significant; ***, p<0.001; one-way ANOVA/Bonferroni's multiple comparisons test. (L) Quantification of the rhabdomere size in in the 30-day-old flies with indicated genotypes. n = 22 for each genotype, data are represented as mean + SD. ns, not significant; ***, p<0.001; one-way ANOVA/Bonferroni's multiple comparisons test.

The online version of this article includes the following source data for figure 1:

**Source data 1.** The numerical data that are represented as a graph in *Figure 1I*.
**Source data 2.** The numerical data that are represented as a graph in *Figure 1J*.
**Source data 3.** The numerical data that are represented as a graph in *Figure 1K*.
**Source data 4.** The numerical data that are represented as a graph in *Figure 1L*.

---

increased the mitochondrial proportion that have contacts with ER (*Figure 2K and Q*). However, it did not significantly change the ERMCS length for each mitochondrion (*Figure 2K and R*). The distance between ER and mitochondria at the ERMCSs in fat body slightly reduced (*Figure 2K and S*). Miga overexpression not only dramatically increased the mitochondrial frequency associated with ER (*Figure 2L and Q*), but also increased the average ERMCS length for each mitochondrion (*Figure 2L and R*). It also greatly reduced the proximity between ER and mitochondria at the ERMCSs (*Figure 2L and S*). Expressing Miga together with Vap33 further increased the ERMCS incidence and length (*Figure 2M and Q–R*) suggesting that Miga and Vap33 together could establish ERMCSs. Vap33 RNAi did not significantly affect ERMCS formation in fly fat body (*Figure 2N and Q–R*). However, ERMCS increase caused by Miga overexpression was canceled by Vap33 RNAi (*Figure 2O and Q–R*). Similarly, Miga^FM overexpression failed to affect ERMCS formation in fat body tissues (*Figure 2P–S*). These data suggest that the interaction between Miga and Vap33 through FFAT motif is critical for establishing ERMCSs. When Miga was overexpressd, we observed enhanced mitochondrial fusion and increased mitochondrial length (*Figure 2—figure supplement 1*; *Zhang et al., 2016*). However, when Miga^FM was overexpressed, the mitochondria length was significantly shorter than that in the wild-type Miga overexpressed tissues (*Figure 2—figure supplement 1*).

## The interaction between Vap33 and Miga is required for Miga overexpression induced neurodegeneration

Since the interaction between Vap33 and Miga is required for establishing ERMCSs, we wondered whether it is also required for the eye degeneration caused by Miga overexpression. We therefore used *GMR-Gal4* to drive Miga^FM expression and examined the eye morphology in 1-day-old animals. Although Miga^FM expressing eyes had some donut shaped mitochondria, they were not closely associated with ER membranes (*Figure 3A'–C'*). Both rhabdomere numbers and shapes were comparable to the controls (*Figure 3A–C, F and G*). Vap33 RNAi alone did not show obvious eye defects (*Figure 3E–E'*, 3F, 3G). Knockdown *Vap33* in the Miga overexpressed retina significantly revived the retinal defects caused by Miga overexpression (*Figure 3D, F and G*). Although there were donut-shaped mitochondria inside the photoreceptor cells, the rhabdomere shapes and sizes were largely normal (*Figure 3D'*). These data suggested that the eye defects caused by Miga overexpression require the interaction between Miga and Vap33.

In addition to eye degeneration, Miga overexpression in muscle also caused flight muscle degeneration in aged (30 days old) flies. However, Miga^FM overexpression only led to very mild muscle degeneration phenotypes in the aged animals (*Figure 3—figure supplement 1*). These data suggested that the interaction between Miga and Vap33 is critical for the defects caused by Miga overexpression.

The interaction between Vap33 and Miga increased ERMCSs and led to eye degeneration. We speculated whether the degeneration was caused by the increasing ERMCSs. In mammals, it has been reported that a protein called PTPIP51 could interact with VAPA/B and mediate ERMCSs (*Stoica et al., 2014*). There is no homolog of PTPIP51 in flies. We expressed the human

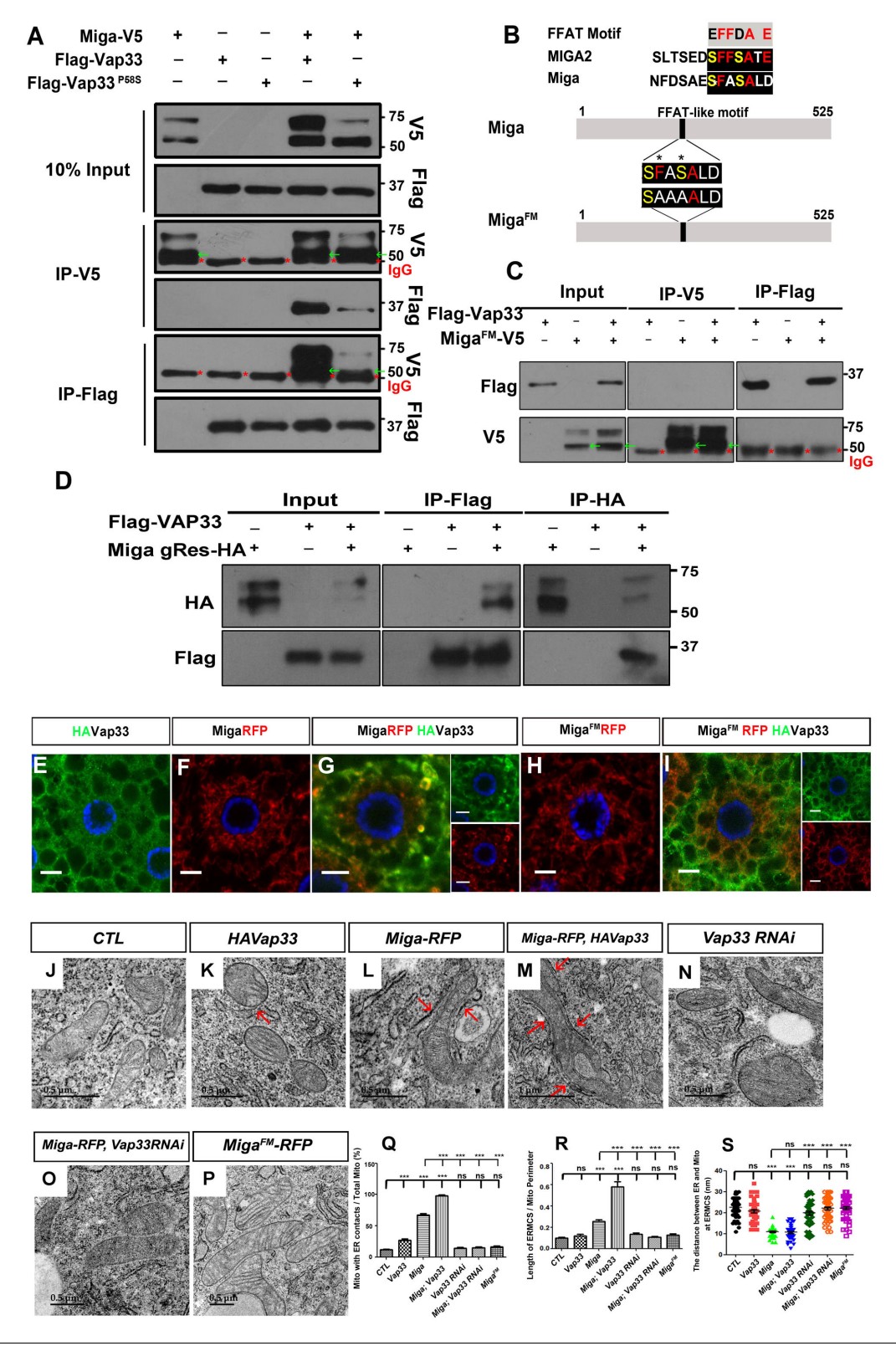

**Figure 2.** Miga forms complex with Vap33 and mediates ERMCSs. (**A**) Miga-V5 and Vap33-Flag could pull down each other in both directions in the IP assay when both were overexpressed in S2 cells. The affinity between Miga-V5 and Vap33^P58S-Flag was reduced compared with that between Miga-V5 and wildtype Vap33-Flag. Miga-V5 showed two bands in most of the blots. The lower bands have a molecular weight close to 50 KD (indicated with green arrows), which were often merged with the IgG heavy chain (indicated with red *) in the blots after IP experiments. (**B**) A scheme to show the

*Figure 2 continued on next page*

*Figure 2 continued*

typical FFAT motif, the FFAT-like motifs in human MIGA2 and *Drosophila* Miga protein, and the amino acid changed in Miga^FM. (C) Miga^FM-V5 and Vap33-Flag failed to pull down each other in the IP assay. Miga^FM-V5 showed two bands in most of the blots. The lower bands have a molecular weight close to 50 KD (indicated with green arrows), which were often merged with the IgG heavy chain (indicated with red *) in the blots after IP experiments. (D) Genomic rescue fragment of Miga with HA tags (Miga gRes-HA, mimics the endogenous Miga expression level ) and Vap33-Flag were expressed in S2 cells and the IP assays were perfromed by IP with anti-HA or IP with anti-Flag antibodies. Miga and Vap33 could pull down each other in this condition. (E–I) HA-tagged Vap33 (green), RFP-tagged Miga or Miga^FM (red) were overexpressed in fat body tissues with indicated combinations. When Vap33 and Miga were co-expressed, they colocalized with each other (G) and the patterns of both proteins were different from the patterns when they were expressed individually (E, F). (H, I) Miga^FM over-expression failed to recruit Vap33. (J–P) TEM of the fat body tissues with indicated genotypes. Miga overexpression increased ERMCSs (L). Co-expression of Miga and Vap33 further increased ERMCS (M). *Vap33* RNAi did not affect ERMCSs (N). Miga overexpression could not induce ERMCS increase when *Vap33* were knock down by RNAi (O). Miga^FM overexpression did not affect ERMCSs (P). The red arrows indicate the ERMCSs. (Q) Quantification of the proportion of mitochondria with ERMCSs. n = 6 images for each genotype. Data are represented as mean + SD. ns, not significant; ***, p<0.001; one-way ANOVA/Bonferroni's multiple comparisons test. (R) Quantification of ratio between the length of ERMCSs and the mitochondrial perimeter. n = 16 for each genotype. Data are represented as mean + SD. ns, not significant; ***, p<0.001; one-way ANOVA/Bonferroni's multiple comparisons test. (S) Quantification of the distance between ER and mitochondria at ERMCSs. n = 50 for each genotype. Data are represented as mean + SD. ns, not significant; ***, p<0.001; one-way ANOVA/Bonferroni's multiple comparisons test.

The online version of this article includes the following source data and figure supplement(s) for figure 2:

**Source data 1.** The numerical data that are represented as a graph in *Figure 2Q*.
**Source data 2.** The numerical data that are represented as a graph in *Figure 2R*.
**Source data 3.** The numerical data that are represented as a graph in *Figure 2S*.
**Source data 4.** The mass spectrometry data to identify the binding partners of Miga.
**Figure supplement 1.** FFAT motif is required for the pro-fusion activity of Miga.
**Figure supplement 1—source data 1.** The numerical data that are represented as a graph in *Figure 2—figure supplement 1*.

PTPIP51 in fly fat bodies and found that the human PTPIP51 is sufficient to increase ERMCS (*Figure 3H–J and L-L''*). We therefore overexpressed the human PTPIP51 and examined the adult fly eyes at days 1 and 30 by TEM. PTPIP51 overexpressed eyes had reduced rhabdomere numbers and sizes from day 1 (*Figure 3O, U and V*). The degeneration was progressively worse with aging. In the 30-day-old flies, most ommatidia lost all rhabdomeres (*Figure 3W*), which is similar to Miga overexpression (*Figure 3R and W*). We also designed a molecular tether with a mitochondrial localization signal from human mitochondrial protein Akap1, a linker, an RFP, and 39 amino acids from the FFAT motif region of Miga (*Figure 3—figure supplement 2A*). The tether interacted with Vap33 nicely (*Figure 3—figure supplement 2B*). When we overexpressed it in fly fat body tissues, we could see ERMCS increase (*Figure 3K*, 3L–L''). When this tether was expressed in the eyes, it also led to reduced rhabdomere numbers and sizes in 1-day-old flies (*Figure 3P, U and V*) and the majority of rhabdomeres were lost in the 30 day old flies (*Figure 3T and W*). These data suggested that artificially increasing ERMCSs led to severe neurodegeneration in fly eyes.

## FFAT motif is important for Miga physiological functions

Since Miga overexpression led to ERMCS increase, we wondered whether Miga was required to establish ERMCSs. We examined the ERMCSs in the fly fat body tissues of wild type and *Miga* mutant third *instar* larvae. However, we did not observe any obvious ERMCS decrease in *Miga* mutants (*Figure 4A–C*). Since ERMCSs were not prevalent in the fat body tissues, the mild reduction was difficult to be detected. This may explain why *Vap33* RNAi also did not affect ERMCSs in the fat body tissues.

Our previous study showed that *Miga* loss led to early pupal fatality and the mosaic eyes with *Miga* mutant clones showed progressive degeneration (*Zhang et al., 2016*). The genomic fragment containing only *Miga* could rescue the fatality and the eye degeneration phenotypes. To test whether the interaction between Miga and Vap33 play any role in the physiological conditions, we made a *Miga* genomic rescue transgene (*Miga^FM gRes HA*) with mutated FFAT motif to ensure that Miga^FM expression levels and patterns mimicked those of endogenous Miga proteins. *Miga^FM* genomic rescue transgene could rescue the *Miga* mutant fatality, but the life span of the *Miga^FM* rescued flies was shorter than those of the ones with wildtype genomic rescue transgene (*Figure 4D*). We also examined the eyes of the genomic rescued adult flies at days 1 and 25. The seventh/eighth photoreceptor cells often degenerated in the 25 day old *Miga^FM* rescued flies, which is barely seen in

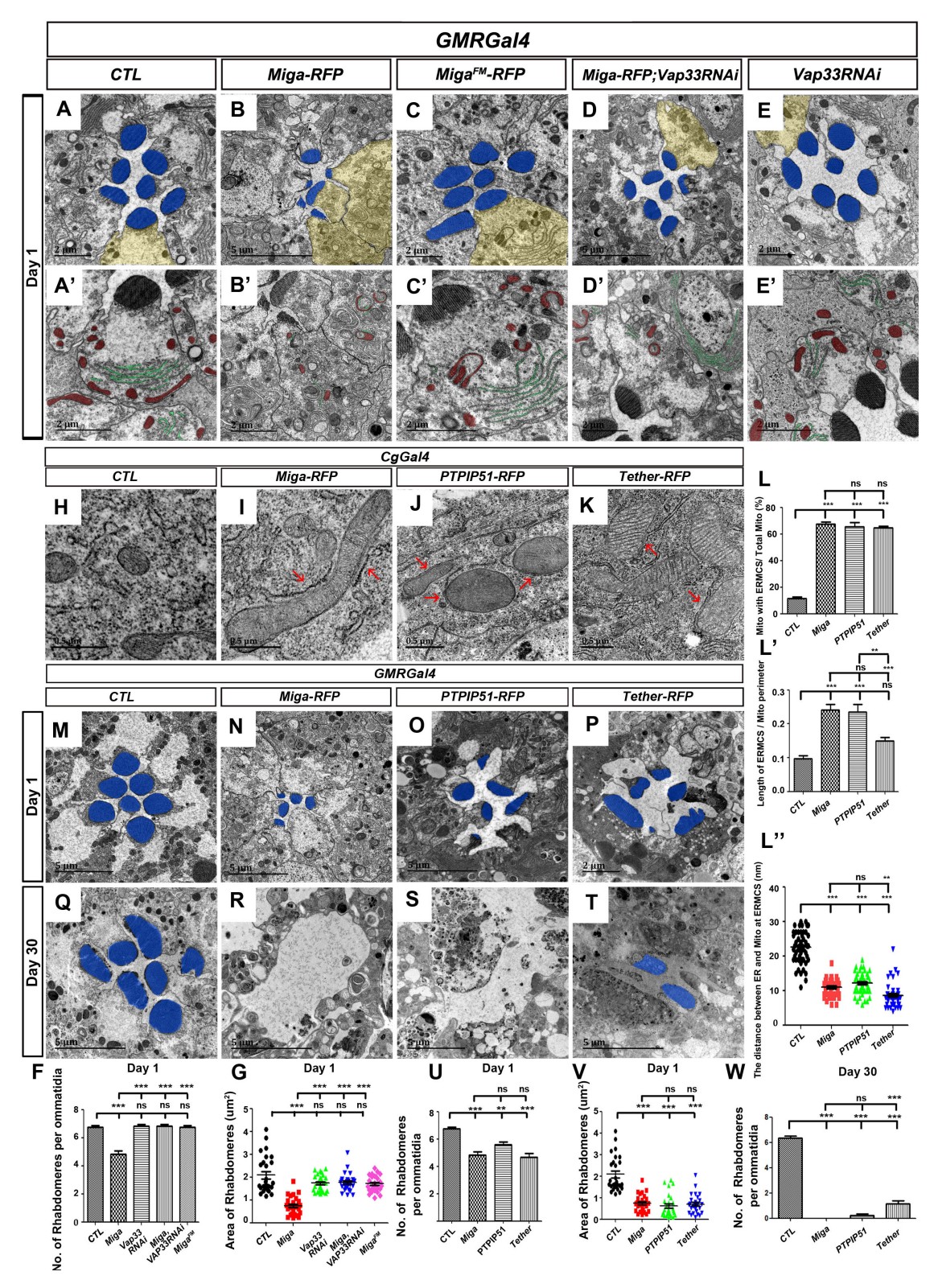

**Figure 3.** Increased ERMCSs led to neurodegeneration. The interaction between Miga and Vap33 is critical for the eye degeneration caused by Miga overexpression. TEM analysis was performed for the retina thin sections of 1-day-old flies with indicated genotypes. (A, A') The ommatidia of the control (CTL) flies showed seven photoreceptor cells with intact rhabdomeres (highlighted with blue pseudo-color). (B, B') *GMR-Gal4* driven Miga-RFP overexpression resulted in reduced number and size of rhabdomeres in the 1-day-old flies. (C, C') *GMR-Gal4*-driven Miga^FM-RFP overexpression did

*Figure 3 continued on next page*

*Figure 3 continued*

not affect the number and size of rhabdomeres in the 1-day-old flies. (D, D') *GMR-Gal4*-driven Miga-RFP overexpression together with Vap33 RNAi did not affect the number and size of rhabdomeres in the 1-day-old flies. (E, E') *GMR-Gal4*-driven Vap33 RNAi did not cause obvious defects in the 1-day-old fly eyes. (A'-E') were enlarged views of the photoreceptor cells highlighted with yellow pseudo-color in (A–E). (F) Quantification of the rhabdomere numbers per ommatidia of fly eyes with indicated genotypes. n = 12 for each genotype, data are represented as mean + SD. ns, not significant; ***, p<0.001; one-way ANOVA/Bonferroni's multiple comparisons test. (G) Quantification of the rhabdomere size of the fly eyes with indicated genotypes. n = 27 for each genotype, data are represented as mean + SD. ns, not significant; ***, p<0.001; one-way ANOVA/Bonferroni's multiple comparisons test. (H–K) TEM analysis was performed for the fat body thin sections of the early third instar larvae with indicated genotypes. Miga-RFP, human PTPIP51-RFP and the artificial ER- mitochondrial tether (Tether-RFP) overexpression led to increase of ERMCS. Red arrows indicate ERMCSs. (L–L'') Quantification of the ERMCSs in the fat body tissues with indicated genotypes. Data are represented as mean + SD. ns, not significant; **, p<0.01, ***, p<0.001; one-way ANOVA/Bonferroni's multiple comparisons test. (M–T) TEM analysis was performed for the retina thin sections of 1-day (M–P) and 30-day-old flies (Q–T) with indicated genotypes. Miga, PTPIP51 and Tether overexpression led to severe retinal degeneration in fly eyes with aging. (U) Quantification of the rhabdomere numbers per ommatidia of the fly eyes with indicated genotypes. n = 12 for each genotype, Data are represented as mean + SD. ns, not significant; **, p<0.01, ***, p<0.001; one-way ANOVA/Bonferroni's multiple comparisons test. (V) Quantification of the rhabdomere size of the fly eyes with indicated genotypes. n = 27 for each genotype, data are represented as mean + SD. ns, not significant; **, p<0.01, ***, p<0.001; one-way ANOVA/Bonferroni's multiple comparisons test. (W) Quantification of the rhabdomere numbers of the fly eyes with indicated genotypes. n = 12 for each genotype, data are represented as mean + SD. ns, not significant; ***, p<0.001; one-way ANOVA/Bonferroni's multiple comparisons test.

The online version of this article includes the following source data and figure supplement(s) for figure 3:

**Source data 1.** The numerical data that are represented as a graph in *Figure 3F*.
**Source data 2.** The numerical data that are represented as a graph in *Figure 3G*.
**Source data 3.** The numerical data that are represented as a graph in *Figure 3L*.
**Source data 4.** The numerical data that are represented as a graph in *Figure 3L'*.
**Source data 5.** The numerical data that are represented as a graph in *Figure 3L''*.
**Source data 6.** The numerical data that are represented as a graph in *Figure 3U*.
**Source data 7.** The numerical data that are represented as a graph in *Figure 3V*.
**Source data 8.** The numerical data that are represented as a graph in *Figure 3W*.
**Figure supplement 1.** FFAT motif are required for the activity of Miga.
**Figure supplement 1—source data 1.** The numerical data that are represented as a graph in *Figure 3—figure supplement 1B*.
**Figure supplement 2.** The artificial ER-mitochondria tether interacts with Vap33.

the wildtype *Miga* rescued flies (*Miga mu; Miga gRes HA*) (*Figure 4E–M*). These data suggested although *Miga* loss did not decrease ERMCSs in fat body tissues, the FFAT motif is critical for Miga physiological functions.

## Miga was phosphorylated at multiple clusters

When we performed the western blot to detect the interaction between Vap33 and Miga, we realized that Miga shows multiple bands and the higher bands have larger molecular weights than the predicted molecular weights (*Figure 2A*). Interestingly, expressing together with Vap33 increased the amount of upper shift bands of Miga (*Figure 2A*), suggesting that Miga modification was regulated.

We then immunoprecipitated Miga from the cultured S2 cells with V5-tagged overexpressed Miga and performed mass spectrometry analysis. Many phosphorylation sites were identified in Miga (*Figure 5A*). To confirm that the upper shift of Miga was indeed due to hyperphosphorylation, we expressed V5-tagged Miga in S2 cells and treated cell lysates with or without λ-phosphatase and examined Miga by western blot with anti-V5 antibody. The phosphatase-treated group showed only one band that run faster than the lowest band in the untreated groups (*Figure 5B*), suggesting that the upper shift of the bands was due to Miga hyperphosphorylation. Due to technical difficulty, one large peptide with multiple serine/threonine (Ser/Thr) was missing in the mass spectrometry analysis. Since hyperphosphorylation usually takes place in clustered sites, we defined seven Ser/Thr clusters and performed the mutagenesis at one cluster per time to examine whether the upper shift of Miga was affected (*Figure 5A,C*). Consistent with the mass spectrometry data, mutated Ser/Thr in clusters I, II, III, and V but not in clusters IV, VI, and VII affected the Miga band shifts (*Figure 5A,C*). Indeed, mutating Ser/Thr to Ala in I, II, and III clusters together abolished the shift of the upper band and mutating Ser/Thr to Glu in these clusters mimic the upper shift of Miga (*Figure 5D*). Mutating Ser/ Thr in cluster V affected both upper and lower band shifts, suggesting that cluster V phosphorylation

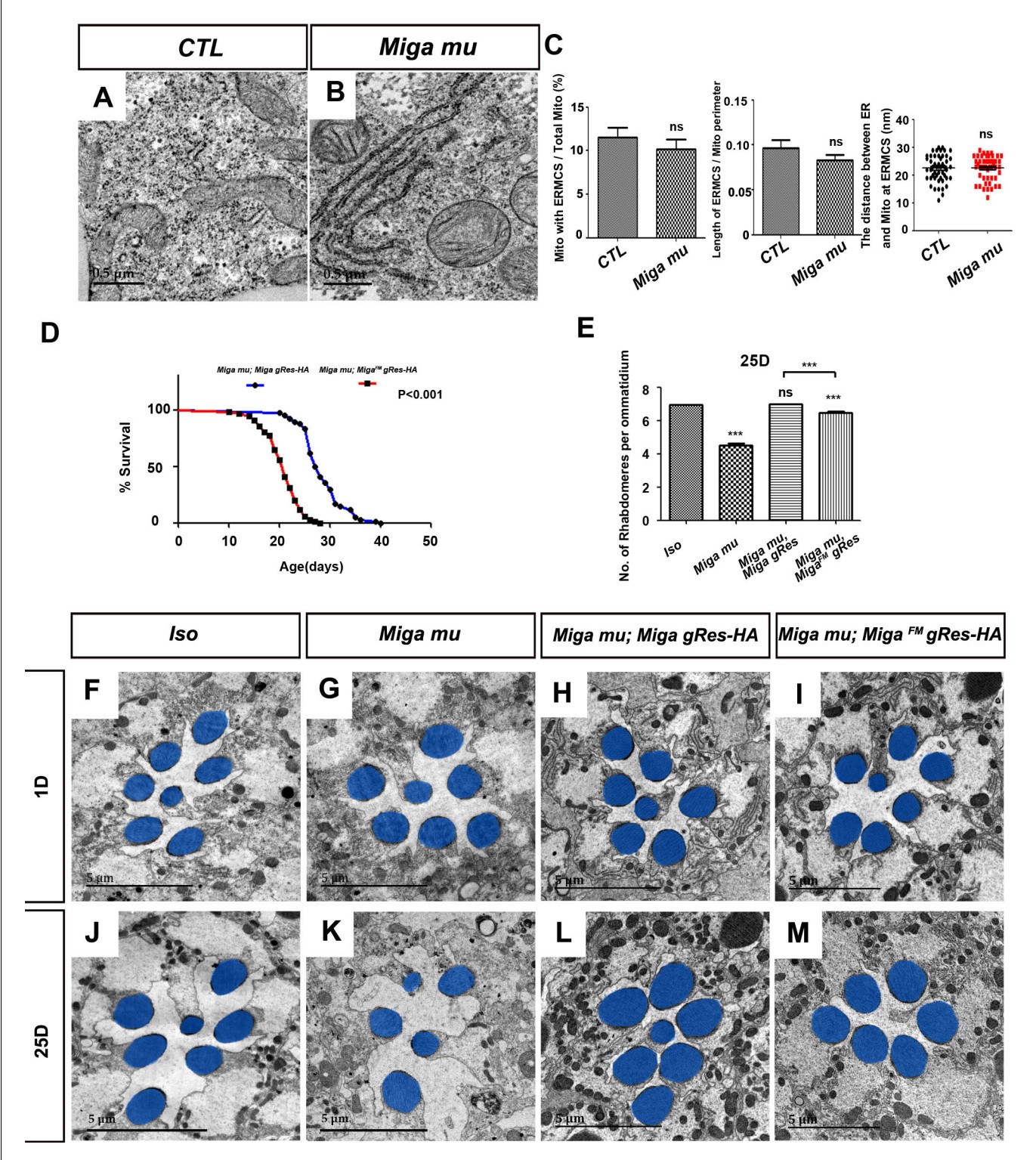

**Figure 4.** FFAT motif is important for the physiological functions of Miga. (**A, B**) TEM analysis was performed for the fat body thin sections of the early third instar larvae with indicated genotypes. The loss of *Miga* did not affect MERCS significantly in fat body tissues. (**C**) Quantification of the ERMCSs in the fat bodies of the indicated genotypes. ns, not significant; two-tailed unpaired t-test. (**D**) The genomic fragment of Miga with FFAT motif mutated (*Miga^FM gRes-HA*) could rescue the fatality of Miga mutants (*Miga mu*), but the rescued flies (*Miga mu; Miga^FM gRes-HA*) have reduced life span compared with *Miga* mutants rescued with wildtype *Miga* genomic fragment (*Miga mu; Miga gRes-HA*). The male flies were analyzed. p<0.001, log rank test; n = 100 flies. (**E**) Quantification of the rhabdomere numbers per ommatidia of the fly eyes with indicated genotypes n = 50 for each genotype,

*Figure 4 continued on next page*

*Figure 4 continued*

data are represented as mean + SD. ns, not significant; ***, p<0.001; one-way ANOVA/Bonferroni's multiple comparisons test. (F–M) TEM analysis was performed for the retina thin sections of 1-day and 25-day-old flies with indicated genotypes. The seventh/eighth photoreceptor cells often degenerated in the 25-day-old *Miga mu; Miga^FM gRes-HA* flies. The rhabdomeres were highlighted with blue pseudo-color.

The online version of this article includes the following source data for figure 4:

**Source data 1.** The numerical data that are represented as a graph in *Figure 4C*.
**Source data 2.** The numerical data that are represented as a graph in *Figure 4D*.
**Source data 3.** The numerical data that are represented as a graph in *Figure 4E*.

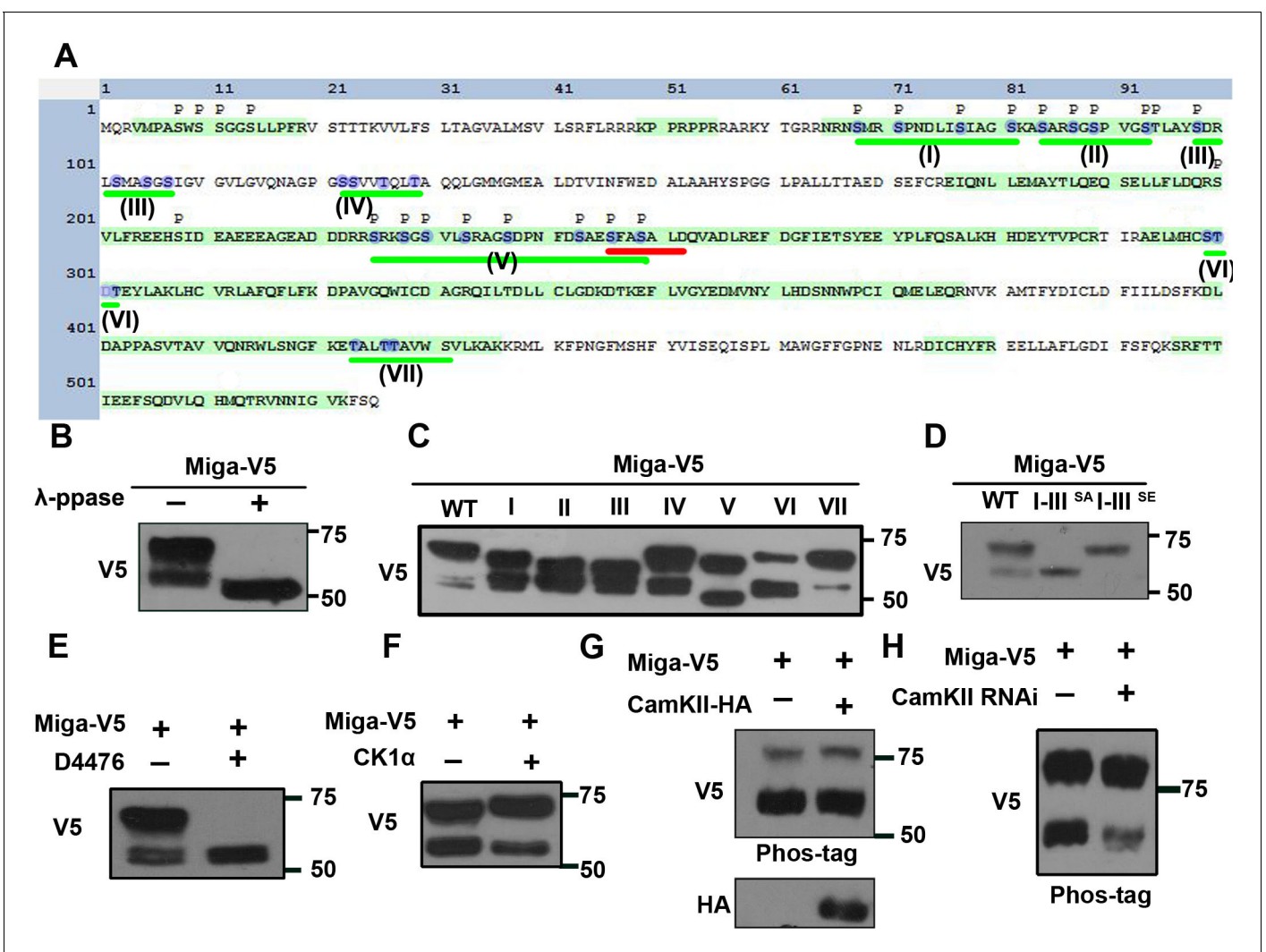

**Figure 5.** Miga was phosphorylated at multiple clusters. (A) A diagram showed the protein sequence of Miga. The sequences highlighted with green color were the peptides identified in the mass spectrometry analysis. The phosphorylation sites identified in the mass spectrometry analysis were labeled with 'p'. The Ser/Thr clusters were underlined with green lines and labeled with I-VII sequentially. The blue color highlighted the amino acid residues were mutated to Ala or Glu in our study. The FFAT motif was underlined with red line. (B) λ-phosphatase treatment abolished the upper shift of V5-tagged Miga protein in the western blot assay. (C) V5-tagged Miga proteins with mutated Ser/Thr in cluster I, II, III, IV, V, VI or VII were analyzed by western blot and the mobility shift of the mutated proteins were compared with the wildtype Miga. (D) Miga proteins with Ser/Thr residues in cluster I-III mutated to Ala (I-III^SA) or Glu (I-III^SE) were analyzed by western blot the mobility shifts of the mutated proteins were compared with the wildtype Miga (WT). (E) CKI inhibitor D4476 inhibited the shift of the upper band of Miga. (F) Coexpressing CKIα with Miga increased the proportion of the upper band vs lower bands. (G) Overexpressed CAMKII together with Miga led to up-shift of both lower and upper bands of Miga in a phospho-tag gel. (H) Phospho-tag gel analysis together with western blot indicated that a slight reduction of band shift when CAMKII was knocked down.

accounted for Miga basal phosphorylation. We then examined which kinases are required for Miga hyperphosphorylation. The inhibitor treatments revealed that the CKI inhibitor D4476 inhibited the Miga upper band shift (*Figure 5E*). The CKI family of serine/threonine protein kinases is involved in regulations of many important signaling pathways and play many critical cellular functions (*Knippschild et al., 2005*). Coexpressing CKIα with Miga increased the upper band proportion vs lower bands (*Figure 5F*). Several Ser residues in cluster V were predicted by the online program GPS (http://gps.biocuckoo.cn/) (*Xue et al., 2011*) and NetPhos 3.1 (http://www.cbs.dtu.dk/services/NetPhos/) (*Blom et al., 2004*) as potential CaMKII phosphorylation sites. CaMKII is a serine/threonine protein kinase that is regulated by the $Ca^{2+}$/calmodulin complex. CaMKII is involved in many signaling cascades and cellular events (*Takemoto-Kimura et al., 2017*). We overexpressed CaMKII along with Miga and examined whether the addition of CaMKII changes the mobility shift of Miga. Considering that the mobility shift might be small, we used phospho-tag gel that increases the mobility shift of the phospho-proteins to detect the band shift of Miga. With co-expression of CaMKII, we observed slight up-shift of both lower and upper bands of Miga (*Figure 5G*). We also detected a slight band shift reduction for both lower and upper bands of Miga when CaMKII was knocked down (*Figure 5H*). These data suggest that CKI and CaMKII are kinases required for Miga phosphorylation.

## Hyperphosphorylation regulates ERMCS formation and fine-tuned Miga activity

Interestingly, Ser246 and Ser249 in cluster V were two phospho-Ser residues located inside the FFAT motif. In the classical FFAT motifs, those two positions were the acidic amino acids E or D. We hypothesized that the phosphorylation of these two serine residues is required for the interaction between Miga and Vap33 and the phosphorylation provides an opportunity to regulate the ERMCS formation. Mutating Ser246 and Ser249 to Ala indeed abolished the interaction between Miga and Vap33 (*Figure 6B*) and failed to increase ERMCSs when it was overexpressed in fly fat bodies (*Figure 6A*). There were eight Ser residues inside the cluster V. Mutating the other six serine residues outside the FFAT motif also greatly reduced the affinity between Miga and Vap33 (*Figure 6—figure supplement 1*). These data suggested that cluster V phosphorylation is required for the interaction between Miga and Vap33.

To evaluate the FFAT motif phosphorylation, we generated an antibody specifically recognizing Ser246 and Ser249 phosphorylation. The antibody was proved to be specific because it only recognized the wild type Miga but not Miga[S246A, S249A] when they were overexpressed in S2 cells (*Figure 6C*).

It has been reported that starvation enhances ERMCS formation (*Yang et al., 2018*). In the wild-type fly fat body, starvation slightly increased the proportion of mitochondria with ERMCSs (*Figure 6H and J*). When Miga was overexpressed in fat body, ERMCSs were greatly increased (*Figure 6G and J–L*). Upon starvation, although there is no further increase of the proportions of mitochondria with ERMCSs, the ERMCS length was increased (*Figure 6I–L*). We wonder whether Miga FFAT motif phosphorylation was increased with starvation. We used Hank's balanced salt solution (HBSS) to treat the cultured S2 cells with V5-tagged overexpressed Miga and examined the Ser246 and Ser249 phosphorylation with western blot at different time points. The Ser246 and Ser249 phosphorylation levels were indeed increased with the HBSS treatments (*Figure 6D–E*). Similarly, our previous study showed that mitochondrial oxidative phosphorylation uncoupler CCCP treatment increased ERMCS (*Yang et al., 2018*). We also detected Ser246 and Ser249 phosphorylation increase in CCCP-treated cells (*Figure 6—figure supplement 2*), although the upper shift of Miga phosphorylation bands was abolished upon this treatment. These data suggested that FFAT motif phosphorylation regulated ERMCS establishment.

In addition to cluster V phosphorylation, Miga was also hyperphosphorylated in the I, II, and, III clusters. To understand how Miga hyperphosphorylation in the first three clusters regulates Miga activity, we overexpressed wild-type Miga, Miga I-III[SA], and Miga I-III[SE] in various tissues and examined their effects. In the larvae fat body tissues, wild-type Miga, Miga I-III[SA] or Miga I-III[SE] overexpression all could increase ERMCSs. There was no obvious difference between these three forms (*Figure 6—figure supplement 3*), suggesting that Miga hyperphosphorylation in these clusters did not affect Miga ability to mediate ERMCSs. Using *GMR-Gal4* to express the three forms of Miga in the developing eyes, we found that all forms cause eye defects and degeneration. Among the three

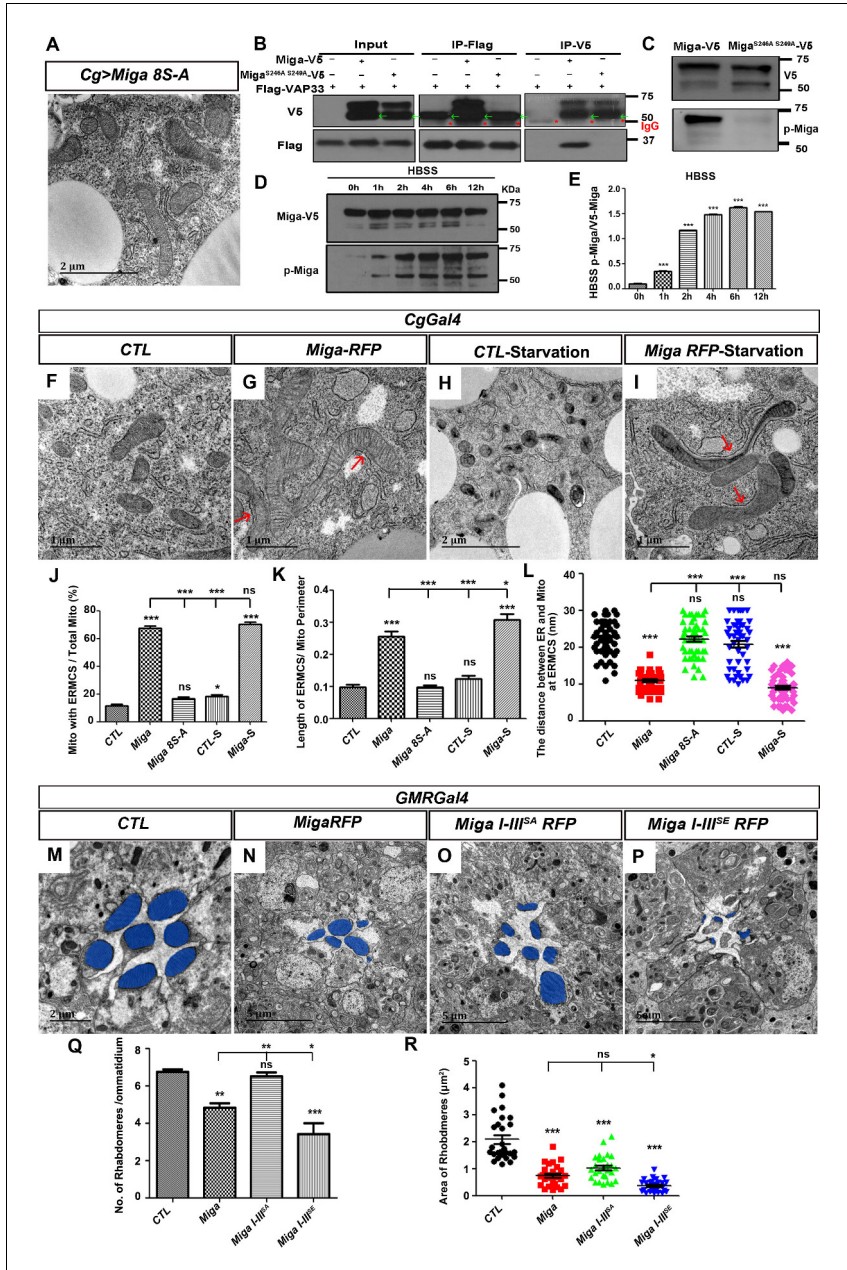

**Figure 6.** Hyper-phosphorylation regulates ERMCS formation and fine-tuned activity of Miga. (**A**) TEM analysis was performed for the fat body thin sections of the early third instar larvae. Overexpression of the mutant form of Miga with Ser residues in the V cluster mutated to Ala (Miga 8S-A) did not affect ERMCSs in fat body tissues. (**B**) Miga with Ser 246 and Ser 249 mutated to Ala (Miga$^{S246A, S249A}$) fail to bind to Vap33. Miga$^{S246A, S249A}$-V5 showed two bands in most of the blots. The lower bands have a molecular weight close to 50 KD (indicated with green arrows), which were often merged with the IgG heavy chain (indicated with red *) in the blots after IP experiments. (**C**) A phospho-specific antibody of Miga (p-Miga) recognize overexpressed wildtype Miga-V5 but not Miga$^{S246A, S249A}$-V5. Western blot with anti-V5 antibody indicated that both proteins were expressed at comparable levels. (**D, E**) HBSS treatment increased the phosphorylation on the Ser246 and Ser249 residues. (**E**) was the quantification of the ratios between p-Miga and total V5 tagged Miga when the cells were treated with HBSS. n = 3. Data are represented as mean + SD. ns, not significant; *, p<0.05, **, p<0.01, ***, p<0.001; one-way ANOVA/Bonferroni's multiple comparisons test. (**F–I**) TEM analysis was performed for the thin sections of fly early third instar larval fat body tissues with indicated genotypes and treatments. Starvation was performed by treating the dissected fat body tissues with HBSS for 6 hr. (**J–L**) Quantification of the ERMCSs in the fat body tissues with indicated genotypes and treatments. Data are represented as mean + SD. ns, not significant; *, p<0.05, **, p<0.01, ***,

*Figure 6 continued on next page*

*Figure 6 continued*

p<0.001; one-way ANOVA/Bonferroni's multiple comparisons test. Starvation slightly increased the proportion of mitochondria with ERMCSs in the control group (CTL-S). When Miga was overexpressed, starvation increased the length of ERMCSs per mitochondria. (**M–P**) TEM analysis was performed for the retina thin sections of 1-day-old flies with indicated genotypes. Overexpression of Miga I-III$^{SE}$ led to more severe eye defects than the overexpression of wild type Miga or Miga I-III$^{SA}$. Overexpression Miga I-III$^{SA}$ had the weakest eye defects when compared with the overexpression of the wildtype Miga or the overexpression of Miga I-III$^{SE}$. The rhabdomeres were highlighted with blue pseudo-color. (**Q**) Quantification of the rhabdomere numbers per ommatidia in the fly eyes with indicated genotypes. n = 12 for each genotype, data are represented as mean + SD. ns, not significant; *, p<0.05, **, p<0.01, ***, p<0.001; one-way ANOVA/Bonferroni's multiple comparisons test. (**R**) Quantification of the rhabdomere size in the fly eyes with indicated genotypes. n = 27 for each genotype, data are represented as mean + SD. ns, not significant; *, p<0.05, ***, p<0.001; one-way ANOVA/Bonferroni's multiple comparisons test. The online version of this article includes the following source data and figure supplement(s) for figure 6:

**Source data 1.** The numerical data that are represented as a graph in *Figure 6E*.
**Source data 2.** The numerical data that are represented as a graph in *Figure 6J*.
**Source data 3.** The numerical data that are represented as a graph in *Figure 6K*.
**Source data 4.** The numerical data that are represented as a graph in *Figure 6L*.
**Source data 5.** The numerical data that are represented as a graph in *Figure 6Q*.
**Source data 6.** The numerical data that are represented as a graph in *Figure 6R*.
**Figure supplement 1.** The phosphorylation on the cluster V Ser residues are critical for the binding between Miga and Vap33.
**Figure supplement 2.** The phosphorylation of Miga was regulated upon stimulation.
**Figure supplement 2—source data 1.** The numerical data that are represented as a graph in *Figure 6—figure supplement 2B*.
**Figure supplement 3.** The phosphorylation of Miga on cluster I-III did not affect its ability to mediate ERMCSs.
**Figure supplement 3—source data 1.** The numerical data that are represented as a graph in *Figure 6—figure supplement 3B*.
**Figure supplement 3—source data 2.** The numerical data that are represented as a graph in *Figure 6—figure supplement 3C*.
**Figure supplement 3—source data 3.** The numerical data that are represented as a graph in *Figure 6—figure supplement 3D*.
**Figure supplement 4.** The phosphorylation on the I-III clusters on Miga increased Miga activity.
**Figure supplement 4—source data 1.** The numerical data that are represented as a graph in *Figure 6—figure supplement 4B*.

---

forms of Miga, Miga I-III$^{SE}$ caused most severe reduction of rhabdomere numbers and size, while Miga I-III$^{SA}$ overexpression led to the mildest defects (*Figure 6M–R*). Similarly, in the adult muscle, Miga I-III$^{SE}$ overexpression led to muscle degeneration and around 50% flight muscles were affected, while only 40% or 20% flight muscles were affected when wild-type Miga or Miga I-III$^{SA}$ were overexpressed (*Figure 6—figure supplement 4*). These data suggested that Miga I-III$^{SE}$ was more active than the wild-type control and Miga I-III$^{SA}$. The hyperphosphorylation in the first three clusters could fine-tune Miga activity.

## The mammalian homolog of Miga, MIGA2, had conserved function in ERMCS formation

Miga has two mammalian homologs, MIGA1 and MIGA2. Both proteins are mitochondrial outer membrane proteins. We then tested whether MIGA1 and MIGA2 could interact with VAPA/B and promote ERMCS formation. Although the FFAT motif was identical in MIGA1 and MIGA2, MIGA1 had low affinity to VAP proteins. We used IP and detected its interaction with VAPA only in one direction (*Figure 7—figure supplement 1*). MIGA2 interacted with both VAPA and VAPB (*Figure 7A–B*). Mutating the FFAT motif in MIGA2 (MIGA2-mu) abolished the interaction between VAPB and MIGA2 (*Figure 7C*). These data were consistent with the results that has been recently reported by Freyre et al. Using COS7 cells, they demonstrated that MIGA2 interacts with both over-expressed and endogenous VAP proteins and the interaction depends on the FFAT motif in MIGA2 and the MSP (Major Sperm Protein) domain in VAP proteins (*Freyre et al., 2019*). When we

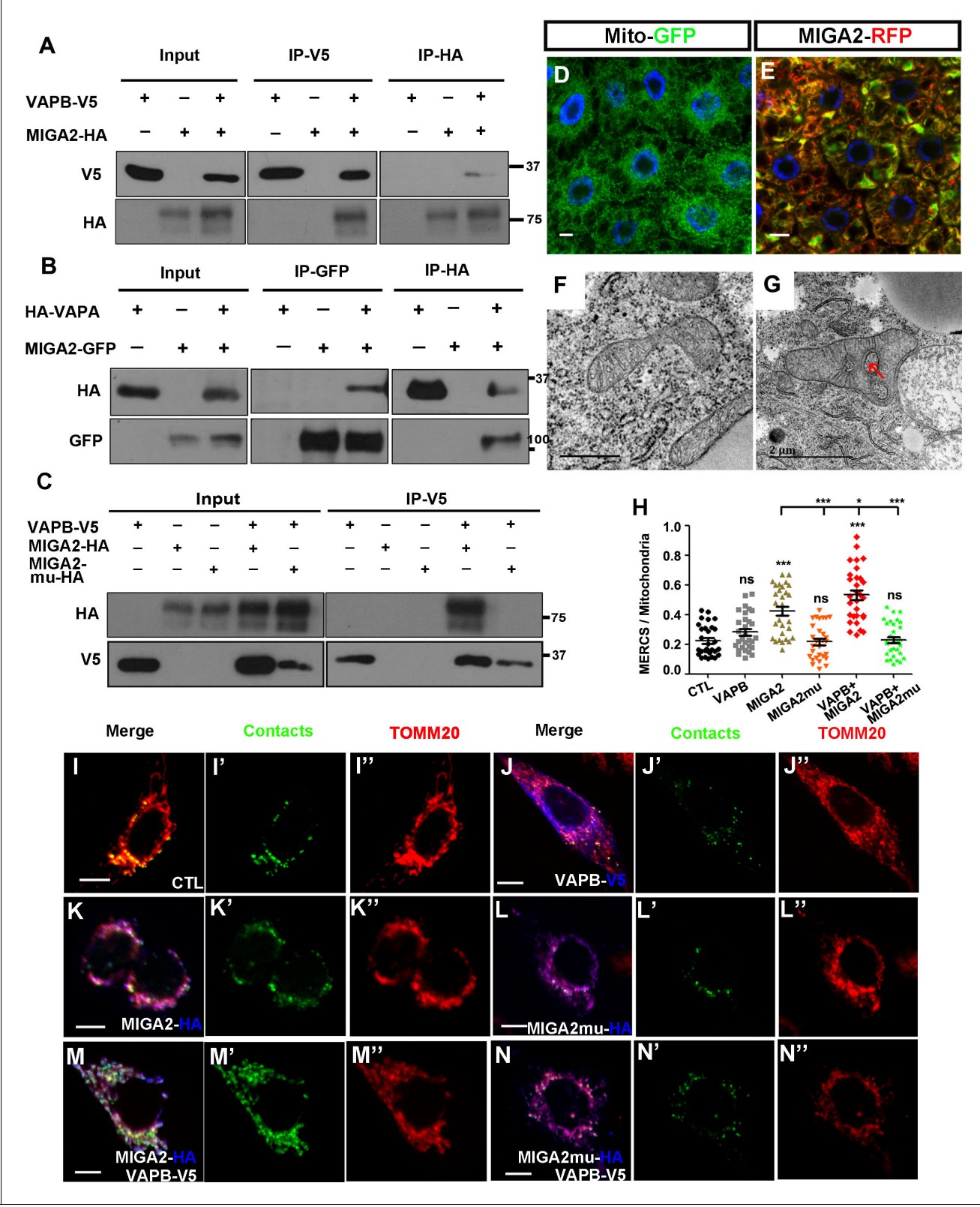

**Figure 7.** The mammalian homolog of Miga, MIGA2, had conserved function in ERMCS formation. (**A**) MIGA2-HA and VAPB-V5 could pull down each other in both directions in the IP assay. (**B**) MIGA2-GFP and VAPA-HA could pull down each other in both directions in the IP assay. (**C**) Mutating the FFAT motif in MIGA2 (MIGA2-mu) abolished the interaction between VAPB and MIGA2 in the IP assay. (**D and E**) Human MIGA2RFP (red) ectopically overexpressed in fly fat body tissues led to the change of MitoGFP (green) patterns. (**F and G**) TEM analysis were performed for the early third instar

*Figure 7 continued on next page*

*Figure 7 continued*

larval fat body tissues with MitoGFP (**F**) or MIGA2RFP together with MitoGFP overexpressed (**G**). MIGA2RFP overexpression increased ERMCS. (**H–N"**) MIGA2 overexpression but not MIGA2-mu overexpression increased the signals of MERCSs. VAPB co-expression with MIGA2 further increased MERCSs. (**H**) Quantification of the ratio between contacts and the total mitochondrial signals of the cells with indicated genotypes. n = 30 for each genotype, ns, not significant; *, p<0.05, ***, p<0.001; one-way ANOVA/Bonferroni's multiple comparisons test. (**I–N"**) A genetic encoded split-GFP based MERCS reporter (green) stably expressed in U2OS cells was used to indicate MERCS. Mitochondria were labeled with anti-TOMM20 staining (red). The expression of VAPB, MIGA2 or MIGA2-mu were indicated by anti-V5 or anti-HA staining (blue).

The online version of this article includes the following source data and figure supplement(s) for figure 7:

**Source data 1.** The numerical data that are represented as a graph in *Figure 7M*.
**Figure supplement 1.** MIGA1 binds to VAPA weakly in HeLa cells when overexpressed.

overexpressed human MIGA2 in fly fat body, mitochondria were swollen and ERMCSs were increased (*Figure 7D–G*).

We previously developed a genetically encoded reporter using split super-folder GFP protein for labeling ERMCSs. MIGA2, but not MIGA2-mu, overexpression increased the ERMCS reporter signals. Co-expression of VAPB with MIGA2, but not MIGA2-mu, further increased the ERMCS reporter signals (*Figure 7H–N"*), suggesting that the interaction between MIGA2 and VAP proteins mediated ERMCSs.

## Discussion

ERMCS mediates the crosstalk between ER and mitochondria. VAP proteins are ER membrane proteins that often serve as an 'entry point' to tether different organelles to ER and mediate the communications between ER and the tethered organelles (*Eisenberg-Bord et al., 2016*). VAP proteins contains the MSP domain that binds to proteins with FFAT motifs. In ALS patients, a point mutation in the VAPB MSP domain was identified (*Nishimura et al., 2004*). The resultant mutant VAPB has reduced affinity to the FFAT motif containing proteins (*Murphy and Levine, 2016*). Indeed, the disease mutation mimic Vap33 had less affinity to Miga than its wild type form. The reduction of organelle contacts including ERMCSs might contribute to the disease conditions in the ALS patients. In this study, we found that increasing ERMCSs by enhancing the interaction between Miga and VAP proteins also led to neurodegeneration, suggesting that the proper amount of contacts and the right distance between ER and mitochondria are critical for neuronal homeostasis.

Although Miga overexpression greatly increased ERMCSs in both fly eyes and fat body tissues, the loss of Miga did not have obvious effects on the ERMCSs in fly fat body tissues. The ERMCSs (10–30 nm), especially the tight contacts (about 10 nm), are rare in the wild type fly fat body. It might be the reason for the difficulty to detect ERMCS reduction in this tissue. We introduced a *Miga* genomic rescue fragment with point mutations in the FFAT motif in *Miga* mutant animals in this study. Although it rescued the fatality of *Miga* mutant, the rescued animals were short lived and had eye degeneration. It suggested that the interaction between Miga and Vap33 had physiological importance. It is interesting that the rescued animals always losing R7/R8 upon aging but not other photoreceptor cells, while the loss of photoreceptor cells in *Miga* mutant animals during aging did not have preference to any particular photoreceptor cells. *Drosophila* has compound eyes composed with repeat units call ommatidia (*Chou et al., 1999*). Each ommatidia consists of eight photoreceptors: the outer photoreceptor cells R1–R6 and the inner photoreceptor cells R7 and R8. The R1–R6 cells express a single opsin Rh1 and are involved in image formation and motion detection. The R7 cells express one of two opsins Rh3 and Rh4. The R8 cells are localized beneath the R7 and express one of the three opsins: Rh3, Rh5, and Rh6. R7 and R8 cells are involved in color vision and polarized light detection. However, we did not observe obvious difference of ERMCSs between the outer and inner photoreceptor cells. Further investigation is required to understand why the FFAT motif in Miga is particularly important for the inner photoreceptor cells. Miga[FM] rescued the lethality of Miga mutants, suggesting that at least some functions of Miga do not need its FFAT motif. The study in mammalian cells indicated that MIGA2 links mitochondria to lipid droplets (LD) through its C-terminal region and is required for adipocyte differentiation. The LD interaction region of MIGA2 is conserved in fly Miga protein. It needs further investigation whether Miga plays a role in lipid metabolism in *Drosophila*.

It has been reported that the ERMCSs marked the mitochondrial fission positions (*Friedman et al., 2011*). However, presenting ERMCS is not sufficient to induce mitochondrial fission. It has been observed that both mitochondrial fission and fusion could happen at ERMCSs (*Guo et al., 2018*). When Miga is overexpressed, we observed increased number of mitochondria with ERMCSs, increased ERMCS length per mitochondrion, and decreased distance between ER and mitochondria at the ERMCSs. Instead of promoting mitochondrial fission, we observed enhanced mitochondrial fusion and dramatically increased mitochondrial length when Miga was overexpressed (*Zhang et al., 2016*). Interestingly, when Miga[FM] was overexpressed, there is no increase of ERMCSs and the mitochondria length was also significantly shorter than that in the wildtype Miga overexpressed tissues (*Figure 2—figure supplement 1*). It has been suggested that Miga function in mitochondrial fusion might be coupled with its function in mediating ERMCSs.

In this study, we found that Miga was hyperphosphorylated at multiple sites. The phosphorylation on the cluster V was essential for the interaction between Miga and VAP protein and the interaction mediated ERMCS establishment. Upon cell stress stimuli, such as starvation, the phosphorylation was increased and so was the ERMCSs. Therefore, the phosphorylation in the cluster V provides a switch to modulate ERMCSs and the subsequent communications between ER and mitochondria. In addition to the cluster V, Miga was also phosphorylated at multiple sites in the clusters I, II, and III. The phosphorylation of these sites was enhanced by the interaction between Miga and Vap33. However, the phosphorylation of these sites did not affect Miga ability to form ERMCSs. Our data indicated that the phosphorylation in the clusters I, II, and III could enhance Miga activity. It would be interesting to know how the phosphorylation affects Miga activity. Although phosphatase treatment abolished the higher molecular weight bands seen in western blot analysis, we cannot exclude that another unknown post-translation modification might also contribute to the molecular weight changes in Miga.

In this study, we identified that CKI and CaMKII were required for Miga phosphorylation. ERMCSs play a key role in calcium flux between ER and mitochondria. The activity of CaMKII is regulated by $Ca^{2+}$/calmodulin. Therefore, the local concentration of calcium might function as a trigger to modulate CaMKII activity and further modify the ERMCSs through the phosphorylation of Miga. We found that the reduction of mobility shift of Miga upon the RNAi of CaMKII was much smaller than that in the cluster V mutants, suggesting that other kinases are involved in the phosphorylation of the cluster V in Miga. We did not perform experiments to test whether CKI or CamKII could modify Miga overexpression phenotypes in fly eyes because of the shutdown of facilities during COVID-19 pandemic. It would be interesting to know whether these kinases are required for Miga overexpression mediated eye degeneration.

It has been reported that presenilins and γ-secretase activity are concentrated in the ERMCSs (*Del Prete et al., 2017*). Increased ERMCSs have also been observed in the fibroblasts from patients with sporadic Alzheimer's disease (*Area-Gomez et al., 2012*). However, whether ERMCS change is the cause or the consequence of the diseases was not known. Our data provided evidence that enhanced ERMCSs by Miga overexpression or other means cause severe degeneration in neurons and muscles. It provided a direct evidence that enhanced ERMCSs is devastating to the cellular homeostasis and leads to neurodegeneration.

# Materials and methods

## Key resources table

| Reagent type (species) or resource | Designation | Source or reference | Identifiers | Additional information |
|---|---|---|---|---|
| Gene (*Drosophila melanogaster*) | Miga | GenBank | FLYB:FBgn0030037 | |
| Gene (*Drosophila melanogaster*) | marf | GenBank | FLYB:FBgn0029870 | |
| Gene (*Drosophila melanogaster*) | MitoPLD | GenBank | FLYB:FBgn0261266 | |

*Continued on next page*

*Continued*

| Reagent type (species) or resource | Designation | Source or reference | Identifiers | Additional information |
|---|---|---|---|---|
| Gene (*Drosophila melanogaster*) | Vap33 | GenBank | FLYB:FBgn0029687 | |
| Gene (*Homo- sapiens*) | Miga1 | GenBank | HGNC:24741 | |
| Gene (*Homo- sapiens*) | PTPIP51 | GenBank | HGNC:25550 | |
| Gene (*Homo- sapiens*) | Miga2 | GenBank | HGNC:23621 | |
| Gene (*Homo- sapiens*) | VAPA | GenBank | HGNC:12648 | |
| Gene (*Homo- sapiens*) | VAPB | GenBank | HGNC:12649 | |
| Cell line (*Drosophila melanogaster*) | S2 | This paper | FLYB:FBtc0000181; RRID:CVCL_Z992 | |
| Cell line (*Homo sapiens*) | U2OS | This paper | CLS Cat# 300364/ p489_U-2_OS, RRID:CVCL_0042 | |
| Cell line (*Homo sapiens*) | Hela | This paper | CLS Cat# 300194/p772_HeLa, RRID:CVCL_0030 | |
| Antibody | anti-HA (Rabbit monoclonal) | Cell Signaling | Cell Signaling Technology Cat# 3724, RRID:AB_1549585 | WB (1:1000) |
| Antibody | anti-GFP (Rabbit polyclonal) | MBL | MBL International Cat# 598, RRID:AB_591819 | WB (1:1000) |
| antibody | anti-Flag (mouse monoclonal) | Sigma | Sigma-Aldrich Cat# F3165, RRID:AB_259529 | WB (1:1000) |
| Antibody | anti-V5 (mouse monoclonal) | Invitrogen | Invitrogen: R96025 | WB(1:5000) IF(1:500) |
| Antibody | P-Miga (Rabbit monoclonal) | GL BioChem Ltd | | WB (1:5000-1:10000) |
| Chemical compound, drug | Phos-tag acrylamide | Boppard | Boppard: 300–93523 | 100 µM |
| Chemical compound, drug | D4470 | Selleck | Selleck:S7642 | 80 µM |
| Chemical compound, drug | CCCP | Sigma | Sigma:C2759 | 10 µM |
| Chemical compound, drug | HBSS | ThermoFisher | Thermo Fisher: 14025076 | |
| Chemical compound, drug | PhosSTOP | Sigma | Sigma: 4906837001 | |
| chemical compound, drug | FLAG peptide | APExBIO | APExBIO: A6001 | |
| Chemical compound, drug | Paraformaldehyde | Electron Microscopy Sciences | Electron Microscopy Sciences: 15711 | 4% |
| Chemical compound, drug | Cacodylic acid | Electron Microscopy Sciences | Electron Microscopy Sciences: 12201 | 1.4% |
| Chemical compound, drug | Glutaraldehyde | Electron Microscopy Sciences | Electron Microscopy Sciences: 16020 | 1% |

*Continued*

| Reagent type (species) or resource | Designation | Source or reference | Identifiers | Additional information |
|---|---|---|---|---|
| Chemical compound, drug | Osmium tetroxide | Electron Microscopy Sciences | Electron Microscopy Sciences: 19152 | 2% |
| Chemical compound, drug | Propylene oxide | Sigma | Sigma:82320 | |
| Chemical compound, drug | Embed 812 | Electron Microscopy Sciences | Electron Microscopy Sciences: 14900 | |
| Chemical compound, drug | DDSA | Electron Microscopy Sciences | Electron Microscopy Sciences: 13710 | |
| Chemical compound, drug | NMA | Electron Microscopy Sciences | Electron Microscopy Sciences:19000 | |
| Chemical compound, drug | DMP-30 | Electron Microscopy Sciences | Electron Microscopy Sciences:13600 | |
| Chemical compound, drug | Uranyl acetate | Electron Microscopy Sciences | Electron Microscopy Sciences: 22400 | 4% |
| Chemical compound, drug | Lead nitrate | Electron Microscopy Sciences | Electron Microscopy Sciences: 17800 | 2.5% |
| Chemical compound, drug | Toluidine blue | Electron Microscopy Sciences | Electron Microscopy Sciences: 22050 | |
| Other | HA beads | Sigma | Sigma:E6779 | |
| Other | Flag beads | Sigma | Sigma, A2220 | |
| Other | Protein A Sepharose 4 Fast Flow beads | GE Healthcare | GE Healthcare: 17-5280-01 | |
| Sequence-based reagent | CaMKII_F | This paper | dsRNA primers | TAATACGACTCACTATAGGGGC AAAGTCCGCTTATTCTCGTTCTT |
| Sequence-based reagent | CaMKII_R | This paper | dsRNA primers | TAATACGACTCACTATAGGGAA TTCTTTGGCTCCCCTCATGC |
| Sequence-based reagent | CaMKII_F | This paper | Real-time PCR primers | ATCCCAACATAGTGCGGCTACATGA' |
| Sequence-based reagent | CaMKII_R | This paper | Real-time PCR primers | AAGTCAGCGAGTTTCACTGCTGCA |

## Molecular cloning

The plasmid pUASattB-RFP was constructed by inserting cDNA of red fluorescent protein (RFP) into pUASattB vector through exogenous restriction sites of KpnI and XbaI. To generate pUASattB Miga-RFP, *Miga* cDNA was cloned into the pUAS attB-RFP vector, RFP was fused to the C terminus of Miga. To generate pUASattB Miga$^{FM}$-RFP, the 247$^{th}$ Phe residue and the 249$^{th}$ Ser residue in Miga were mutated to Ala through site directed mutagenesis. pUASattBMiga$^{S246A, S249A}$-RFP was generated through site directed mutagenesis to mutate the 246$^{th}$ Ser and 249$^{th}$ Ser residues to Ala. pUASattB Miga I-III$^{SA}$-RFP was generated by site directed mutagenesis to change the 68$^{th}$, 71$^{st}$, 77$^{th}$, 81$^{st}$, 84$^{th,}$ 87$^{th}$, 89$^{th}$, 93$^{rd}$, 98$^{th}$, 102$^{nd}$, 105$^{th}$, and 107$^{th}$ Ser residues to Ala. pUASattB Miga I-III$^{SE}$-RFP was generated by site directed mutagenesis to change the 68$^{th}$, 71$^{st}$, 77$^{th}$, 81$^{st}$, 84$^{th,}$ 87$^{th}$, 89$^{th}$, 93$^{rd}$, 98$^{th}$, 102$^{nd}$, 105$^{th}$, and 107$^{th}$ Ser residues to Glu. pUASattB Miga 8SA-RFP was generated by site directed mutagenesis to change the 225$^{th}$, 228$^{th}$, 230$^{th}$, 233$^{rd}$, 237$^{th}$, 243$^{rd}$, 246$^{th}$, and 249$^{th}$ Ser residues to Ala. pattB Miga gRes-HA was generated by cloning Miga genomic DNA with its upstream and downstream 1.5 kb sequence into pattB vector, 3xHA tag was inserted just before the stop codon in the coding sequence (*Zhang et al., 2016*). pattB Miga$^{FM}$ gRes-HA was generated by introduce point mutations to change the 247$^{th}$ Phe residue and the 249$^{th}$ Ser residue to Ala in the *Miga* genomic DNA fragments and subcloned into pattB vector. The genomic DNA fragment

including *Miga*'s gene span together with upstream and downstream 1.5 kb sequences and 3xHA tag was inserted just before the stop codon in the coding sequence. pUASattB PTPIP51-RFP was generated by cloning the full length human *PTPIP51* cDNA into pUASattB-RFP vector. To generate pAC-Miga-V5 plasmid, *Miga* cDNA was sub-cloned into the pAC-His-V5 vector (Invitrogen), V5 was fused to the C terminus of Miga. pAC Miga I-V5 was generated by site-directed mutagenesis to change the 68th, 71st, 77th, 81st Ser residues to Ala. pAC Miga II-V5 was generated by site-directed mutagenesis to change the 84th, 87th, 89th, 93rd Ser residues to Ala. pAC Miga III-V5 was generated by site-directed mutagenesis to change the 98th, 102nd, 105th, and 107th Ser residues to Ala. pAC Miga IV-V5 was generated by site-directed mutagenesis to change the 122nd, 123rd Ser residues to Ala and the 126th, 129th Thr residues to Ala. pAC Miga V-V5 was generated by site-directed mutagenesis to change the 225th, 228th, 230th, 233rd, 237th, 243rd, 246th, and 249th Ser residues to Ala. pAC Miga VI-V5 was generated by site-directed mutagenesis to change the 299th Ser residue to Ala, 300th, 302nd Thr residues to Ala and 301st Asp residue to Ala. pAC Miga VII-V5 was generated by site-directed mutagenesis to change the 423rd, 426th, 427th Thr residues to Ala and 431st Ser residue to Ala. pAC Miga I-III$^{SA}$-V5 was generated by site-directed mutagenesis to change the 68th, 71st, 77th, 81st, 84th, 87th, 89th, 93rd, 98th, 102nd, 105th, and 107th Ser residues to Ala. pAC Miga I-III$^{SE}$-V5 was generated by site-directed mutagenesis to change the 68th, 71st, 77th, 81st, 84th, 87th, 89th, 93rd, 98th, 102nd, 105th, and 107th Ser residues to Glu. pAC-Miga 6S-A-V5 was generated by site-directed mutagenesis to change the 225th, 228th, 230th, 233rd, 237th, 243rd Ser residues to Ala. pAC-Miga 8S-A-V5 was generated by site-directed mutagenesis to change the 225th, 228th, 230th, 233rd, 237th, 243rd, 246th and 249th Ser residues to Ala. pUASVap33-FLAG, pUAS Vap33$^{P58S}$-FLAG, pUAS CK1α have been described before (*Tsuda et al., 2008*; *Jia et al., 2004*). The plasmid pUASattB CamkII-HA was generated by cloning the full length *CamKII* cDNA into pUASattB vector with a N-terminal 3xHA tag. The plasmid pAC-Tether-V5 was generated by cloning the mitochondrial targeting sequence of human mitochondrial protein Akap1 (MAIQFRSLFPLALPGMLALLGWWWFFSRKK), a linker sequence (AEAAAKEAAAKEAAAKA), an *RFP* full length cDNA, and 39 amino acids sequence from FFAT motif region (229aa-267aa) of Miga into pAC-His-V5 vector. pUASattB-Tether was generated by cloning the mitochondrial targeting sequence of human mitochondrial protein Akap1 (MAIQFRSLFPLALPGMLALLGWWWFFSRKK), a linker sequence (AEAAAKEAAAKEAAAKA), an *RFP* full length cDNA, and 39 amino acids sequence from FFAT motif region (229aa-267aa) of Miga into pUASattB vector. pUASattB Miga-FLAG-HA was generated by cloning the full length *Miga* cDNA into pUASattB vector with a FLAG tag (DYKDDDDK) and a HA tag (YPYDVPDYA) fused to the C-terminal of Miga. The plasmid pcDNA3.1-MIGA2-HA was generated by cloning *Miga2* cDNA into pcDNA3.1 vector with a 3xHA tag fused to the C-terminus. pcDNA3.1-MIGA2 mu-HA was generated by site-directed mutagenesis to change the 293rd, 294th Phe residues of MIGA2 to Ala . The plasmid pcDNA3.1-MIGA2-EGFP was generated by cloning *Miga2* cDNA into pcDNA3.1 vector with a C-terminal EGFP tag. The plasmid pcDNA3.1-VAPB-V5 was generated by cloning *VAPB* cDNA into pcDNA3.1 vector with a C-terminal V5 tag. The plasmid pcDNA3.1-VAPA-HA was generated by cloning *VAPA* cDNA into pcDNA3.1 vector with a N-terminal 3xHA tag.

## Fly strains

The fly strains used in this study were listed in the *Supplementary file 1*. All the transgenic strains were generated by the standard way. All the transgenic flies were generated by PhiC31-mediated transgenesis to integrate the DNA fragments at specific sites in the genome. Therefore, the transgenes presumably are single copy insertions.

## Cell culture, transfection and treatments

S2 cells were originally from Invitrogen. HeLa cells and the U2OS cells were originally from ATCC. These cells were recently authenticated and tested for contamination. S2 cells were cultured in a 25°C incubator in Schneider insect cell culture medium (Sigma, S0146). HeLa cells and the U2OS cell lines stable transfected with ERMCS reporters were cultured in a 37°C incubator with 5% $CO_2$ in Dulbecco's modified Eagle's medium (Life Technologies, C11995500CP). Both the culture mediums were supplemented with 10% fetal bovine serum (Life Technologies, 10091148) and 50 IU/mL penicillin/streptomycin (Biological Industries, 03-031-1B). Plasmids were transfected by using lipo2000

(Life Technologies, 11668019) as a transfection reagent. 48 hr after transfection, the cells were harvested and used for immunoprecipitation assay, western blotting or immuno-staining.

For drug treatment, 80 μM CKI inhibitor D4470 (Selleck, S7642) was added into the medium for 48 hr and DMSO was added as a control. For HBSS treatment, cells were cultured in HBSS (Thermo-Fisher, 14025076) for indicated period before harvest. For CCCP (Sigma, C2759) treatment, 10 μM CCCP was added to the culture medium for indicated period before harvest. For λ-ppase treatment, the cells were harvested and lysed in lysis buffer (150 mM NaCl, 50 mM Tris-HCl, 0.5% NP-40, protease inhibitor, pH 8.0) and incubated with λ-ppase for 1 hr at 30℃ before western blot analysis.

## RNA interference (RNAi) in *Drosophila* S2 cells

*Drosophila* RNAi experiments were carried out as previously described. The dsRNA targeting CaM-KII was generated by annealing reverse complimented RNA strains generated by in vitro transcription from a template amplified by PCR using following primers: CaMKII-F: 5'-TAATACGACTCACTATAGGGGCAAAGTCCGCTTATTCTCGTTCTT-3'; CaMKII-R: 5'-TAATACGACTCACTATAGGGAATTCTTTGGCTCCCCTCATGC-3'. dsRNA was transfected to the S2 cells with the liposome RNAi max (Life Technologies, 13778 150). After 3 days, cells were transfected with pAC-Miga-V5. After 2 more days, cells were collected for real-time quantitative PCR and western blot. The primers used for the real time PCR are: CamKII-F: 5'-ATCCCAACATAGTGCGGCTACATGA-3'; CamKII-R: 5'-AAGTCAGC-GAGTTTCACTGCTGCA-3'. *RpLP0*-F: 5'-CTAAGCTGTCGCACAAATGGC-3', *RpLP0*-R: 5'-ATCTCCTTGCGCTTCTTGGA-3'.

## Immunofluorescence

The fat body tissues or the cultured cells were fixed in 4% paraformaldehyde (Sigma-Aldrich, 158127) for 45 min, followed by permeabilizing in PBST (PBS with 0.1% Triton X-100 (Sangon Biotech, T0694)). Samples were incubated with primary antibody at 4℃ overnight. After washing with PBST, samples were then incubated with secondary antibodies for 1 hr at room temperature in dark. After that, samples were mounted in 80% glycerol (Sangon Biotech, A100854) with 5 ng/μL DAPI (Invitrogen, D-1306) followed by confocal microscopy (Cal Zeiss, LSM710, Oberkochen, Germany).

## TEM analysis

For the adult fly eyes, fly head was dissected and fixed in EM eye solutions (1.4% cacodylic acid (Electron Microscopy Sciences,12201), 4% paraformaldehyde (Electron Microscopy Sciences, 15711), and 1% Glutaraldehyde (Electron Microscopy Sciences,16020), pH7.2.) for more than 48 hr at 4℃. Samples were then washed five times with Millipore water, and post-fixed with 2% osmium tetroxide (Electron Microscopy Sciences, 19152) for about 2 hr. After rinsing five times with Millipore water, samples were gradually dehydrated through a graded series of ethanol (50%, 70%, 80%, 90%, 95%, and 100%, respectively). After that, samples were dehydrated in propylene oxide (PO) (Sigma, 82320) for three times with 30 min for each time. And then, samples were embedded in Eponate 12 resin, which was made up from Embed 812 (Electron Microscopy Sciences, 14900), DDSA (Electron Microscopy Sciences, 13710), NMA (Electron Microscopy Sciences,19000) and DMP-30 (Electron Microscopy Sciences, 13600). The samples were then cured in 65℃ for 48 hr. For the fat body and the muscle tissues, the initial fixation solution is 2.5% glutaraldehyde (Electron Microscopy Sciences, 16020) and the samples were washed with PBS after fixation. All the other steps were same as those used for the fly eyes. The samples were cut into 50 nm thin sections and stained with 4% uranyl acetate (Electron Microscopy Sciences, 22400) and 2.5% lead nitrate (Electron Microscopy Sciences, 17800) for electron microscopy analysis (Hitachi Ltd., HT7700, Tokyo, Japan). For the semi-thin sections of muscle tissues, the samples were cut into 1.5 μm thin sections and stained with toluidine blue (Electron Microscopy Sciences, 22050) and examine under a light microscopy (Nikon Corporation, ECLIPSE 80i, Tokyo, Japan).

## Co-immunoprecipitation and western blotting

Cells were lysed in lysis buffer (150 mM NaCl, 50 mM Tris-HCl pH 8.0,1 mM EDTA, 0.5% Triton-100 (for HeLa cell); 150 mM NaCl,50 mM Tris-HCl pH 8.0, 10 mM NaF, 1 mM $Na_3VO_4$,1% NP-40, 10% glycerol, 1.5 mM EDTA pH 8.0 (for S2 cell)), supplemented with protease inhibitors PMSF, aprotinin, pepstatin, leupeptin, and phosphatase inhibitor PhosSTOP (sigma, 4906837001). Samples were

centrifuged at 16000 g for 10 min and supernatant were collected and incubated with HA beads (Sigma, E6779) or Flag beads (Sigma, A2220) or V5 antibody followed with Protein A Sepharose 4 Fast Flow beads (GE Healthcare, 17-5280-01) for 2–4 hr at 4°C. Spin down the beats at 500 g for 30 s. Then washed with lysis buffer for three times and add SDS running buffer to the beads and proceed to western blot analysis. For western blotting analysis, proteins were separated by SDS–PAGE, and transferred onto a PVDF membrane. The membrane was then blocked with 5% non-fat milk (Sangon Biotech, A600669) in TBST buffer and incubated with primary antibodies in TBST with 5% non-fat milk overnight at 4°C. The membranes were then washed in TBST and incubated with HRP labeled secondary antibodies (1:5000 in TBST with 5% non-fat milk) for 1 hr at RT. The membranes were then washed in TBST and developed with ECL reagents (Cyanagen Srl, XLS3-0020) and exposed. Quantification of protein bands was done with Image J software.

## Phos-tag SDS-PAGE

Phos-tag SDS-PAGE was performed with 7% polyacrylamide gels containing 100 µM Phos-tag acrylamide (Boppard, 300–93523) and 200 µM $MnCl_2$. After electrophoresis, Phos-tag acrylamide gels were washed with transfer buffer (50 mM Tris, 384 mM glycine, 0.1% SDS, 20% methanol) containing 10 mM EDTA for 20 min and with transfer buffer without EDTA for 10 min. The proteins on the gel was transferred onto PVDF membranes followed with regular western blot procedure.

## Antibodies

Anti-HA antibody (Cell Signaling: #3724) was used with 1:1000 dilution in both western blot and immunofluorescence staining. Anti-GFP antibody (MBL: 598) was used with 1:1000 dilution in western blot. Anti-Flag antibody (Sigma: F3165) was used with 1:1000 dilution in western blot. Anti-V5 antibody (Invitrogen: R96025) was used with 1:5000 dilution in western blot and 1:500 dilution in immunofluorescence staining. Miga phospho-specific antibody was generated by GL BioChem Ltd. Phosphorylated peptide GSDPNFDSAE(S)pFA(S)pA was used as an antigen to immune the rabbits. The antibody was tested by ELESA assays and western blot (1:5000).

## Mass spectrometry

To identify Miga's binding proteins, FLAG-HA tagged Miga was overexpressed in S2 cells for 48 hr. Cells were lysed in lysis buffer (150 mM NaCl, 50 mM Tris-HCl pH 8.0, 10 mM NaF, 1 mM $Na_3VO_4$, 10% NP-40, 10% glycerol, 1.5 mM EDTA pH 8.0) and Miga was pulled down by FLAG beads and eluted with FLAG peptide (APExBIO, A6001) twice. The eluted fractions then subjected for pull-down assay with HA beads. Then the pull-down products were separated by SDS-PAGE. The gel was dyed with Coomassie brilliant blue for 30 min and then de-colored by de-staining solution (75% alcohol, acetic acid, and $H_2O$). Then the gel was cut and subjected to the LC-MS/MS analysis.

To identify the modifications on Miga protein, V5 tagged Miga was overexpressed in S2 cells for 48 hr. Cells were lysed in lysis buffer and IP with V5 beats. The proteins were separated by SDS-PAGE and dyed by Coomassie brilliant blue for 30 min and then incubated with de-staining solution (75% alcohol, acetic acid, and $H_2O$). The gel corresponding to the size of Miga (50–75 KDa) was cut. In gel digestion was performed with trypsin at 37°C overnight. Peptides were extracted with 50% acetonitrile/ 5% formic acid, followed by 100% acetonitrile. Peptides were dried to completion and resuspended in 2% acetonitrile/ 0.1% formic acid. The tryptic peptides were dissolved in 0.1% formic acid (solvent A), directly loaded onto a home-made reversed-phase analytical column (15 cm length, 75 µm i.d.). The gradient was comprised of an increase from 6% to 23% solvent B (0.1% formic acid in 98% acetonitrile) over 16 min, 23% to 35% in 8 min and climbing to 80% in 3 min then holding at 80% for the last 3 min, all at a constant flow rate of 400 nl/min on an EASY-nLC 1000 UPLC system.

The peptides were subjected to NSI source followed by tandem mass spectrometry (MS/MS) in Q Exactive Plus (Thermo) coupled online to the UPLC. The electrospray voltage applied was 2.0 kV. The m/z scan range was 350 to 1800 for full scan, and intact peptides were detected in the Orbitrap at a resolution of 70,000. Peptides were then selected for MS/MS using NCE setting as 28 and the fragments were detected in the Orbitrap at a resolution of 17,500. A data-dependent procedure that alternated between one MS scan followed by 20 MS/MS scans with 15.0 s dynamic exclusion. Automatic gain control (AGC) was set at 5E4. The resulting MS/MS data were processed using Proteome Discoverer 1.3. Tandem mass spectra were searched against Miga protein sequence. Trypsin/

P was specified as cleavage enzyme allowing up to two missing cleavages. Mass error was set to 10 ppm for precursor ions and 0.02 Da for fragment ions. Phosphorylation of serine, threonine and tyrosine were specified as fixed modification and oxidation on Met was specified as variable modifications. Peptide confidence was set at high, and peptide ion score was set >20.

### Lifespan analysis

The *Drosophila* of each genotype were collected and housed at a density of 10 flies per vial (n = 100). All flies were kept in a constant temperature and humidity environment with 12 hr on/off light cycle. The number of surviving animals was counted and transferred to fresh food every 2 days.

## Acknowledgements

We are grateful to THFC, BDSC, and DGRC for providing fly strains and cDNA clones. Dr. Tong is supported by National Natural Science Foundation of China (91754103, 31622034, 31571383), National Key Research & Developmental Program of China (2017YFC1001100, 2017YFC1001500), Natural Science Foundation of Zhejiang Province, China (LR16C070001), Fundamental research funds for the central universities. Dr. Tong is a Qianjiang Scholar. We are also grateful for the technical supports provided by Dr. Bing Yang, LSI Mass Spectrometry Facility, and Imaging Core.

## Additional information

### Funding

| Funder | Grant reference number | Author |
| --- | --- | --- |
| National Natural Science Foundation of China | 91754103 | Chao Tong |
| National Natural Science Foundation of China | 31622034 | Chao Tong |
| National Natural Science Foundation of China | 31571383 | Chao Tong |
| National key research and developmental program of China | 2017YFC1001100 | Chao Tong |
| National key research and developmental program of China | 2017YFC1001500 | Chao Tong |
| Natural Science Foundation of Zhejiang Province | LR16C070001 | Chao Tong |

The funders had no role in study design, data collection and interpretation, or the decision to submit the work for publication.

### Author contributions

Lingna Xu, Xi Wang, Data curation, Formal analysis, Validation, Investigation, Visualization, Methodology, Writing - original draft; Jia Zhou, Data curation, Formal analysis, Validation, Investigation, Visualization; Yunyi Qiu, Data curation, Formal analysis, Investigation, Methodology; Weina Shang, Investigation, Methodology; Jun-Ping Liu, Liquan Wang, Resources, Investigation, Project administration; Chao Tong, Conceptualization, Resources, Data curation, Formal analysis, Supervision, Funding acquisition, Validation, Investigation, Methodology, Writing - original draft, Project administration, Writing - review and editing

### Author ORCIDs

Lingna Xu https://orcid.org/0000-0002-0534-8781
Chao Tong https://orcid.org/0000-0001-6521-5465

### Decision letter and Author response

Decision letter https://doi.org/10.7554/eLife.56584.sa1
Author response https://doi.org/10.7554/eLife.56584.sa2

## Additional files

### Supplementary files
• Supplementary file 1. The genotypes of the fly strains used in this study.

• Transparent reporting form

### Data availability
All data generated or analyzed during this study are included in the manuscript and supporting files.

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
