## [Decision Letter]

**Acceptance summary:**

Your timely discovery of the interaction between the mitochondrial protein Miga and the endoplasmic reticulum protein Vap33 in *Drosophila* is in line with the very recent analogous discovery of the Klemm group in mammals. Moreover, your discovery of the phosphoregulation of this binding, where phosphate group additions compensate for the poorly acidic nature of Miga's "di-phenylalanine in acidic track" (FFAT) motif sheds a new light in the regulation of FFAT motifs, which play pleiotropic roles in organelle interactions.

**Decision letter after peer review:**

Thank you for submitting your article "Miga mediated endoplasmic reticulum-mitochondria contact sites regulate neuronal homeostasis" for consideration by *eLife*. Your article has been reviewed by three peer reviewers, including Benoit Kornmann as the Reviewing Editor and Reviewer #1, and the evaluation has been overseen by Suzanne Pfeffer as the Senior Editor.

The reviewers have discussed the reviews with one another and the Reviewing Editor has drafted this decision to help you prepare a revised submission.

Summary:

In the present manuscript, Xu et al. Identify Vap33, an ER protein, as a binding partner for Miga, a mitochondrial protein. They show that vap33 binds Miga through a FFAT motif. When overexpressing Miga, this causes a massive increase in ER mitochondria contacts, which is dependent upon binding to Vap33. This increase in contacts, whether it is caused by Miga overexpression of by more artificial means (expression of a mammalian or completely artificial tethering protein) is toxic for omatidial development. The authors also show that while loss of miga is lethal, the lethality can be complemented by the expression of a Miga mutant form unable to bind vap33, indicating that Miga's essential role is not via Vap33 binding and the promotion of ER-mitochondria contacts. Moreover, lack of Miga or Vap33 has little impact on existing ER-mitochondria contacts. Finally, the authors show that Miga's FFAT (two phenylalanines in acidic tract) motif is unconventional, in that it is not really acidic, but can be made acidic through phosphorylation of serine residues, thus tuning affinity to Vap33.

As you will read from the attached reviews, the works was deemed interesting, yet insufficiently supported at places. In particular, most of the data here are from overexpression experiments. The interaction of Vap33 and Miga is not shown for proteins expressed at endogenous levels and the physiological implication of overexpressing the proteins are not straightforward. In particular, the ALS causing mutations are thought to reduce vapb binding to FFAT motifs, while the overexpression herein would rather increase it. Moreover, the text is at places incomplete or imprecise. In particular, the study of Freire et al., 2020, which shows interaction with mammalian Miga-2 and VAP and predates the present study by a few months should be much more thoroughly cited.

Revisions expected in follow-up work:

Pull down should be attempted with endogenously expressed proteins.

Reviewer #1

This is carefully executed and exciting story but its main problem is that it comes after Freyre et al., 2019, who already demonstrate binding of VAP and miga in cultured mammalian cells. Given the novelty of the Freyre publication and the need for "scooping protection", this need not preclude publication. However, it is important that this previous work is properly acknowledged along the whole manuscript and not as a side note in the Introduction. Freyre et al. already demonstrated VAP-Miga binding, the dependency upon the degenerated FFAT motif, the effect in increasing ER-mitochondria contacts by Miga overexpression, the relocalization of VAP to mitochondria upon miga overexpression.

Additional point:

– Miga has at least one function that doesn't require Vap33 binding, since lethality of a miga mutant can be rescued by expressing a FFAT-less Miga. What is/are this/ese function(s)? Do they relate to mitochondrial fusion. Is there any genetic interaction between Miga and mitoPLD or Marf to support this? A discussion of the other function(s) of Miga would be useful.

Reviewer #2

The manuscript by Xu and Wang et al. reports on mitoguardin (MIGA) mediated interaction between the ER and mitochondria and a critical function in maintenance of the retina. The results presented here confirm published work on the mammalian ortholog MIGA2, and add interesting new data on the activation of MIGA by phosphorylation. The function of ERMCs in neuronal homeostasis are also highly relevant, and the insights will be of high interest to the general readership of *eLife*. This reviewer recommends publication after the following points have been addressed.

The novelty of the paper relates to the characterization of Miga function in neurodegeneration (e.g. Figure 1, and 4), and the characterization of Miga activation by phosphorylation (Figure 5).

For a general reader it is not trivial to recognize the cell types and sub-cellular structures described in Figure 1. It would be helpful to segment and label the rhabdomeres, as well as indicate the outline of mitochondria and the ER in all panels that are compared. The areas enlarged in the figure panels called e.g. A', B' etc. should be outlined in the corresponding panels showing lower magnification, e.g. A, B etc. This also applies for the panels in Figure 3.

It would be helpful to include an introductory sentence to the kinases studied in Figure 5, spell out CK and CamKII when used first, and explain what is known about their function, possibly even already in the Introduction. While the inhibitor experiments in Figure 5 are very convincing it is not clear why the WT condition in Figure 5C and the untreated condition in Figure 5F (which should be the same conditions) are showing almost the opposite band intensities between non-phosphorylated and phosphorylated Miga-V5. This reviewer is also not confident in agreeing with the claimed effect of CamKII. The data in Figure 5G and H rather indicate that CamKII RNAi increases MIGA phosphorylation. The unphosphorylated bands in the phostag gels seem less intense than in controls. Consider changing the text to explain this figure panels better, or interpret the data differently.

The interaction with the VAP proteins was recently reported in work on mammalian MIGA2, and this work should be cited in the section reporting the interaction between Vap33 and Miga. This also applies for the Introduction and the text to Figure 2 and Figure 7. It is a strength of this paper to show that the molecular properties of mitoguardins are evolutionarily conserved and the authors might want to highlight this fact instead of ignoring it.

The text to Figure 7A is confusing and does not match the labelling of the figure panels. E.g. the interaction between MIGA2 and VAPA and VAPB is shown in the panel, and the text indicates that MIGA1 is shown. The text jumps to Figure 7N before the rest of the panels are described. Consider the relevant corrections and changes in the text.

The reason for the changes in relative abundance of full length MIGA-V5 and p-MIGA shown in Figure 6B are not clear. Are the authors suggesting that phosphorylation of the p-clusters I-III occur independently of starvation and only p-cluster V is modified after starvation? Or is this increase in phosphorylation contributed only from endogenous Miga. It seems important to show that the increase of p-Miga in Figure 6G is inhibited after the addition of the kinase inhibitors.

VAP proteins are tail anchored proteins and C-terminal tagging might be problematic in particular when using a FLAG tag which is highly charged. This could explain the issues with the Co-IP experiments reported in Figure 7. Have the authors considered using N-terminally tagged.

The Materials and methods section indicates the use of 10% NP-40 in the lysis and IP buffer used for mass spectrometry proteomics. Please double check the amount of detergent and correct the text if needed.

Reviewer #3

Xu et al., argue that Miga interact with the Vap33 protein through its FFAT-like motif and regulates ERMCSs. They argue that this interaction and elevated ERMCSs can promote photoreceptor degeneration. They also show that Miga can be regulated through phosphorylation and this phosphorylation is required for Vap33 binding as well ERMCSs formation.

Overall, this manuscript is well structured with interesting discovery on the function and regulation of Miga in ERMCSs and neurodegeneration. A major concern is that much of the work is based overexpression data but it can be considered for publication after addressing my concerns below:

1) Figure 2D-H,

a) Please show a separate channel for Figure 2F and H. It is difficult to compare expression patterns with merged channels.

b) Figure 2D-F, co-expression of Vap33 and Miga leads to a different expression pattern of both proteins, indicating the interaction/co-localization of the two protein may be due to an artifact from over-expression. Please try IP/IF with endogenous proteins (such as Miga gRES HA, mentioned later in the manuscript)

c) If the Miga-Vap33 protein interaction is weak at basal level, then how significant is this interaction in the cell. This is an important question that needs to be addressed.

2) The authors argue that Miga changes its expression pattern upon Vap33 co-expression: "many circular structures appeared". However, there are already some circular structures with Miga expression alone (Figure 2E). More representative images and proper quantification are required to make this statement.

3) Figure 2J: The authors argue that: Vap33 overexpression alone increased the mitochondrial proportion that have contacts with ER. However, I don't see a mitochondria-ER contact in this image. Please label where the contact is.

4) As Vap33 is linked to ALS, does Vap33-RNAi also suppresses the flight muscle degeneration in Miga over-expression flies?

5) Vap33-P58S, is associated with ALS. However, Vap33-P58S variant impairs the interaction with FFAT motif containing proteins (such as OSBP), suggesting that loss of Vap33-FFAT interaction is associated with ALS. However, in this manuscript, the authors argue that increased Vap33-Miga interaction lead to neurodegeneration. They should discuss this. Does overexpression of Vap33-P58S variant modify the photoreceptor loss in Miga over-expressing flies?

6) Subsection “Miga was phosphorylated at multiple clusters”. Please explain how CKI inhibitor D4476 is identified.

7) In the same section the authors claim that: Several Ser residues in cluster V were predicted as potential CaMKII phosphorylation sites. Please specify how phosphorylation sites prediction is performed.

8) As CKI and CaMKII regulates Miga phosphorylation. And the author argue that this phosphorylation is required for ERMCSs formation. Can CKI and CaMKII modify the photoreceptor defect observed in Miga overexpression flies?

---

## [Author Response]

Revisions for this paper:As you will read from the attached reviews, the works was deemed interesting, yet insufficiently supported at places. In particular, most of the data here are from overexpression experiments. The interaction of Vap33 and Miga is not shown for proteins expressed at endogenous levels and the physiological implication of overexpressing the proteins are not straightforward. In particular, the ALS causing mutations are thought to reduce vapb binding to FFAT motifs, while the overexpression herein would rather increase it. Moreover, the text is at places incomplete or imprecise. In particular, the study of Freire et al., 2020, which shows interaction with mammalian Miga-2 and VAP and predates the present study by a few months should be much more thoroughly cited.Revisions expected in follow-up work:Pull down should be attempted with endogenously expressed proteins.

During the revision process, we made several attempts to address this problem. We had two anti-Miga antibodies that we generated in this study, but none of them worked properly in immunoprecipitation. We then expressed a genomic rescue construct of Miga with HA tag fused to its C-terminus, which likely expressed in an endogenous level. We do not have anti- VAP33 antibody and fail to obtain from colleagues in other country because of the outbreak of COVID-19 and shut down of many facilities. we overexpressed Flag tagged VAP33 and performed IP experiments. Since VAP33 overexpression only marginally increased ERMCSs in our experimental settings, we think the overexpression might not be very problematic. We included these data in the revised manuscript (Figure 2D). Although the experimental condition is not ideal, that is the best we can do now.

Reviewer #1This is carefully executed and exciting story but its main problem is that it comes after Freyre et al., 2019, who already demonstrate binding of VAP and miga in cultured mammalian cells. Given the novelty of the Freyre publication and the need for "scooping protection", this need not preclude publication. However, it is important that this previous work is properly acknowledged along the whole manuscript and not as a side note in the Introduction. Freyre et al. already demonstrated VAP-Miga binding, the dependency upon the degenerated FFAT motif, the effect in increasing ER-mitochondria contacts by Miga overexpression, the relocalization of VAP to mitochondria upon miga overexpression.

Thanks for the comments. In the revised manuscript, we thoroughly acknowledged the

previous publication.

Additional point:– Miga has at least one function that doesn't require Vap33 binding, since lethality of a miga mutant can be rescued by expressing a FFAT-less Miga. What is/are this/ese function(s)? Do they relate to mitochondrial fusion. Is there any genetic interaction between Miga and mitoPLD or Marf to support this? A discussion of the other function(s) of Miga would be useful.

We did not test the genetic interaction between the FFAT-less Miga and MitoPLD or Marf. In this study, we found that when Miga^FM^ was overexpressed, there is no increase of ERMCSs and the mitochondria length was also significantly shorter than that in the wildtype Miga overexpressed tissues (Figure 2—figure supplement 1). It suggested that Miga functions in mitochondrial fusion might be coupled with its function in mediating ERMCSs. The study in mammalian cells indicated that MIGA2 links mitochondria to lipid droplets (LD) through its C-terminal region and is required for adipocyte differentiation. The LD interaction region of MIGA2 is conserved in fly Miga protein. It needs further investigation whether Miga plays a role in lipid metabolism in *Drosophila*. We add discussion regarding to this point in the revised manuscript.

Reviewer #2The manuscript by Xu and Wang et al. reports on mitoguardin (MIGA) mediated interaction between the ER and mitochondria and a critical function in maintenance of the retina. The results presented here confirm published work on the mammalian ortholog MIGA2, and add interesting new data on the activation of MIGA by phosphorylation. The function of ERMCs in neuronal homeostasis are also highly relevant, and the insights will be of high interest to the general readership of eLife. This reviewer recommends publication after the following points have been addressed.The novelty of the paper relates to the characterization of Miga function in neurodegeneration (e.g. Figure 1, and 4), and the characterization of Miga activation by phosphorylation (Figure 5).For a general reader it is not trivial to recognize the cell types and sub-cellular structures described in Figure 1. It would be helpful to segment and label the rhabdomeres, as well as indicate the outline of mitochondria and the ER in all panels that are compared. The areas enlarged in the figure panels called e.g. A', B' etc. should be outlined in the corresponding panels showing lower magnification, e.g. A, B etc. This also applies for the panels in Figure 3.

Thanks for the advice. We made the changes accordingly.

It would be helpful to include an introductory sentence to the kinases studied in Figure 5, spell out CK and CamKII when used first, and explain what is known about their function, possibly even already in the Introduction. While the inhibitor experiments in Figure 5 are very convincing it is not clear why the WT condition in Figure 5C and the untreated condition in Figure 5F (which should be the same conditions) are showing almost the opposite band intensities between non-phosphorylated and phosphorylated Miga-V5. This reviewer is also not confident in agreeing with the claimed effect of CamKII. The data in Figure 5G and H rather indicate that CamKII RNAi increases MIGA phosphorylation. The unphosphorylated bands in the phostag gels seem less intense than in controls. Consider changing the text to explain this figure panels better, or interpret the data differently.

The introduction of the kinases was added in the revised manuscript. The difference between the original Figure 5C and Figure 5F was because of the different sample preparation procedures. The samples in original 5C were prepared by adding SDS loading buffer directly to the pelleted cells. The samples in original Figure 5F were prepared by using lysis buffer to treat cells for 1hour, centrifuging for 10 minutes with high speed, collecting 20 uL supernatant and then adding SDS loading buffer (the rest of the supernatant was collected for IP experiments). The long preparation procedure for original Figure 5F likely reduced the phosphorylation in the protein samples. Now we changed Figure 5F with a new image in which the samples were prepared in the same way as Figure 5C. Since the phosphatase activity is very high in the lysates and the p-Miga is often de-phosphorylated. It is meaningful to compare the bands intensity and shift for the samples prepared side by side but not between different experiments.

Figure 5G and H were phostag gel, the band shift in both images indicating CamKII might contribute to the phosphorylation of Miga. The overall intensity was low in the CamKII RNAi groups. We repeated the experiment several times, the RNAi group always have lower expression of Miga (both higher bands and lower bands). It needs further investigation to know why the expression level of Miga was low in the CaMKII RNAi group. Never the less, the mobility shift of both the lower bands and the upper bands was slower than the controls, indicating CaMKII contributes to the mobility shift in the control cells. We changed the image of Figure 5H with a better exposure time to show both bands of Miga. We also made some changes in the text and included discussions about the kinases in the revised manuscript.

The interaction with the VAP proteins was recently reported in work on mammalian MIGA2, and this work should be cited in the section reporting the interaction between Vap33 and Miga. This also applies for the Introduction and the text to Figure 2 and Figure 7. It is a strength of this paper to show that the molecular properties of mitoguardins are evolutionarily conserved and the authors might want to highlight this fact instead of ignoring it.

Thanks for the comments. We acknowledged the recent publication of Miga thoroughly in the revised manuscript.

The text to Figure 7A is confusing and does not match the labelling of the figure panels. E.g. the interaction between MIGA2 and VAPA and VAPB is shown in the panel, and the text indicates that MIGA1 is shown. The text jumps to Figure 7N before the rest of the panels are described. Consider the relevant corrections and changes in the text.

Sorry for the confusion. We added the MIGA1 IP results in the new supplemental figure (Figure 7—figure supplement 1). We also reorganized the Figure 7 to make the flow better.

The reason for the changes in relative abundance of full length MIGA-V5 and p-MIGA shown in Figure 6B are not clear. Are the authors suggesting that phosphorylation of the p-clusters I-III occur independently of starvation and only p-cluster V is modified after starvation? Or is this increase in phosphorylation contributed only from endogenous Miga. It seems important to show that the increase of p-Miga in Figure 6G is inhibited after the addition of the kinase inhibitors.

We prefer the first possibility suggested by the reviewer. We tried to detect the phosphorylation of endogenous Miga with this antibody and fail to obtained positive signals with or without HBSS treatments. Therefore, the p-Miga signals detected in MIga-V5 expression samples likely is the phosphorylated Miga-V5 protein. RNAi CAMKII did not change the level of p-Miga significantly. We think CAMKII might phosphorylate cluster V on the other sites rather than the S246 and S249 in the FFAT motif recognized by the phosph-specific antibody of Miga. We believe that there are other kinases also responsible for Miga’s phosphorylation. We discussed this point in the revised manuscript.

VAP proteins are tail anchored proteins and C-terminal tagging might be problematic in particular when using a FLAG tag which is highly charged. This could explain the issues with the Co-IP experiments reported in Figure 7. Have the authors considered using N-terminally tagged.

It is a good suggestion. The Vap33 vector with Flag tag was tagged at the N-terminus. And the VAPA vector was also tagged at the N-terminus. The VAPB vector used here was tagged with V5 tag and it is tagged at the C-terminus. It has normal ER distribution, forms complex with MIGA2, and increases ERMCSs when co-expressed with MIGA2, suggesting VAPB-V5 functions normal. To prevent the confusion of the tag positions, we re-labeled the VAP proteins as Flag-Vap33, VAPA-HA, and V5-VAPB in the revised figures.

The Materials and methods section indicates the use of 10% NP-40 in the lysis and IP buffer used for mass spectrometry proteomics. Please double check the amount of detergent and correct the text if needed.

It is not correct. We changed it.

Reviewer #3Xu et al., argue that Miga interact with the Vap33 protein through its FFAT-like motif and regulates ERMCSs. They argue that this interaction and elevated ERMCSs can promote photoreceptor degeneration. They also show that Miga can be regulated through phosphorylation and this phosphorylation is required for Vap33 binding as well ERMCSs formation.Overall, this manuscript is well structured with interesting discovery on the function and regulation of Miga in ERMCSs and neurodegeneration. A major concern is that much of the work is based overexpression data but it can be considered for publication after addressing my concerns below:1) Figure 2D-H,a) Please show a separate channel for Figure 2F and H. It is difficult to compare expression patterns with merged channels.

Thanks for the comments. We split the channels in the revised manuscript as suggested.

b) Figure 2D-F, co-expression of Vap33 and Miga leads to a different expression pattern of both proteins, indicating the interaction/co-localization of the two protein may be due to an artifact from over-expression. Please try IP/IF with endogenous proteins (such as Miga gRES HA, mentioned later in the manuscript)

During the revision process, we made several attempts to address this problem. We had two anti-Miga antibodies we generated in this study, but none of them worked properly in immunoprecipitation. We then expressed a genomic rescue construct of Miga with HA tag fused to its C-terminus as suggested by reviewer 2, which likely expressed in an endogenous level. We do not have anti- VAP33 antibody and fail to obtain from colleagues in other country because of the outbreak of COVID-19 and shut down of many facilities. Since VAP33 overexpression only slightly increase ERMCSs in our experimental settings, we overexpressed Flag tagged VAP33 and performed IP experiments. We included these data in the revised manuscript. Although the experimental condition is not ideal, that is the best we can do now.

c) If the Miga-Vap33 protein interaction is weak at basal level, then how significant is this interaction in the cell. This is an important question that needs to be addressed.

As we showed in Figure 4D-M, a genomic rescue fragment of Miga with FFAT motif mutated cannot fully rescue the Miga mutant phenotypes. The rescued flies have reduced life span and retinal degeneration. These data suggested that the interaction between Miga-Vap33 indeed has physiological significance.

2) The authors argue that Miga changes its expression pattern upon Vap33 co-expression: "many circular structures appeared". However, there are already some circular structures with Miga expression alone (Figure 2E). More representative images and proper quantification are required to make this statement.

The circular structures might not be a precise description of the structures we are trying to describe. We removed this statement in the revised manuscript.

3) Figure 2J: The authors argue that: Vap33 overexpression alone increased the mitochondrial proportion that have contacts with ER. However, I don't see a mitochondria-ER contact in this image. Please label where the contact is.

The increase of the mitochondrial proportion that have contacts with ER was modest when Vap33 was overexpression alone. The conclusion was drawing from statistic measurements and it might be hard to see from a single image. We changed an image with ERMCSs and label it with arrow in the revised manuscript.

4) As Vap33 is linked to ALS, does Vap33-RNAi also suppresses the flight muscle degeneration in Miga over-expression flies?

We did not perform this experiment due to the shutdown of EM facility on the campus during the epidemics of COVID-19. It has been reported before that the loss of Vap33 cause severe muscle defects. It is difficult to predict whether Vap33-RNAi could rescue Miga-overexpression phenotypes in muscles.

5) Vap33-P58S, is associated with ALS. However, Vap33-P58S variant impairs the interaction with FFAT motif containing proteins (such as OSBP), suggesting that loss of Vap33-FFAT interaction is associated with ALS. However, in this manuscript, the authors argue that increased Vap33-Miga interaction lead to neurodegeneration. They should discuss this. Does overexpression of Vap33-P58S variant modify the photoreceptor loss in Miga over-expressing flies?

We believe both loss and gain of ERMCS is devastating to the neuronal homeostasis. We discussed this in the manuscript as following “In ALS patients, a point mutation in the VAPB MSP domain was identified. The resultant mutant VAPB has reduced affinity to the FFAT motif containing proteins ). Indeed, the disease mutation mimic Vap33 had less affinity to Miga than its wild type form. The reduction of organelle contacts including ERMCSs might contribute to the disease conditions in the ALS patients.” In the revised manuscript, we further emphasized that both reduction and increase of ERMCS are devastating to the cellular homeostasis and leads to neurodegeneration.

For a similar reason described above, we did not perform the experiments suggested here.

6) Subsection “Miga was phosphorylated at multiple clusters”. Please explain how CKI inhibitor D4476 is identified.

We did a small-scale kinase inhibitor screening with the screen well^R^ kinase inhibitor library (BML-2832 Version 2.2) and found that an inhibitor called 5-Iodotubericidin could inhibit the band shift. However, this inhibitor is not very specific. It inhibits ERK2, adenosine kinase, CKI, CKII, we then tried several other inhibitors including D4476 and found that only this one can inhibit the band shift nicely. During the screening, we also have some other hits that cannot be confirmed by further experiments. It is not very surprising, since most of the inhibitors in this library are not very specific. To simplify the story and make reader easy to follow, we only described D4476 in the manuscript.

7) In the same section the authors claim that: Several Ser residues in cluster V were predicted as potential CaMKII phosphorylation sites. Please specify how phosphorylation sites prediction is performed.

We used online program GPS (http://gps.biocuckoo.cn/) and NetPhos 3.1 (http://www.cbs.dtu.dk/services/NetPhos/) did the prediction. We add the description in the revised manuscript.

8) As CKI and CaMKII regulates Miga phosphorylation. And the author argue that this phosphorylation is required for ERMCSs formation. Can CKI and CaMKII modify the photoreceptor defect observed in Miga overexpression flies?

Because the shutdown of the EM facility, we could not perform the experiments requested here. It is hard to predict whether the knockdown of these kinases will modify the photoreceptor defects in Miga overexpressed flies because these kinases have many different substrates and CKI is important for several signaling pathways for eye development.